


# A quantitative decoupling analysis (QDA v1.0) method for assessing the contributions of meteorology, emissions, and chemistry to fine particulate pollution

Junhua Wang[1,3], Baozhu Ge[*,1,3], Xueshun Chen[1,3], Jie Li[1,3], Keding Lu[2], Yayuan Dong[1,3], Lei Kong[1,3], Zifa Wang[*,1,3], Yuanhang Zhang[2]

[1]State Key Laboratory of Atmospheric Boundary Layer Physics and Atmospheric Chemistry (LAPC), Institute of Atmospheric Physics (IAP), Chinese Academy of Sciences (CAS), Beijing 100029, China
[2]College of Environmental Sciences and Engineering, Peking University, Beijing, 100871, China
[3]College of Earth and Planetary Sciences, University of Chinese Academy of Sciences, Beijing, 100049, China

*Correspondence to*: Baozhu Ge (gebz@mail.iap.ac.cn) and Zifa Wang (zifawang@mail.iap.ac.cn)

**Abstract.** A comprehensive understanding of the effects of meteorology, emissions, and chemistry on severe haze is critical in the mitigation of air pollution. However, such an understanding is greatly hindered by the nonlinearity of atmospheric systems. In this study, we developed the quantitative decoupling analysis (QDA) method to quantify the effects of emissions, meteorology, chemical reactions, and their nonlinear interactions on fine particulate matter ($PM_{2.5}$) pollution by running built-in scenario simulations in each model step. Different from previous methods, the QDA method achieves a fully decomposed analysis of hourly changes in the $PM_{2.5}$ concentration during pollution events into seven parts, including the pure meteorological contribution (M), the pure emissions contribution (E), the pure chemistry contribution (C), and the interactions among these processes (i.e., ME, MC, EC, and MCE). Via embedding the QDA method into the Weather Research and Forecasting–Nested Air Quality Prediction Modeling System, we employed this method and combined it with the Integrated Process Rate method to study a typical heavy haze episode in Beijing. We evaluate the model performance against *in situ* meteorological and air quality observations and describe the QDA analytical factors of this case. Results showed that M varied most significantly at different stages of the episode, from 0.21 $\mu g \cdot m^{-3} \cdot h^{-1}$ during the accumulation stage to $-11.82$ $\mu g \cdot m^{-3} \cdot h^{-1}$ during the removal stage, indicating that the pure meteorological contribution dominated the hourly fluctuation amplitude of the $PM_{2.5}$ concentration. M acted as the most important cleaner for $PM_{2.5}$ in non-polluting periods but stopped being effective at this and instead became a contributor in the accumulation stage such that $PM_{2.5}$ tended to grow rapidly under the superimposed influence of emissions and chemical processes, which would probably mark the beginning of a heavy pollution event. The contribution of E ranged from 0.63 to 0.88 $\mu g \cdot m^{-3} \cdot h^{-1}$ owing to the diurnal variation of emissions. The pure chemical contribution was shown to increase with the level of haze, becoming the largest (0.37 $\mu g \cdot m^{-3} \cdot h^{-1}$) in the maintenance period, which was 25% higher than during the pre-contamination period. And C+CE made a significant contribution in the accumulation and maintenance stages, indicating that chemical reactions are more important in the polluted period than in other periods. Nonnegligible nonlinear effects exist among the processes of meteorology, emissions, and chemistry on $PM_{2.5}$ concentrations ($-1.83$ to 2.44 $\mu g \cdot m^{-3} \cdot h^{-1}$)—something that has generally been ignored in previous studies and during the





development of heavy-pollution control strategies. The nonlinear effects are helpful in eliminating the interference of other
processes and obtaining a more purified result of the target process and have important indicative significances. The ratio of
CE to C is positively correlated with the chemical speed. For precursors like $NH_3$, the smaller value of CE in the most polluted
period indicated that $NH_3$ was more deficient, and thus reducing emissions of it in that period would have had the most efficient
controlling effect on the $PM_{2.5}$. This study highlights that the QDA method can be used to realize an in-depth understanding
of the effects of adverse meteorological conditions in haze and to judge whether the precursors are excessive or not. Not only
can the QDA method provide researchers and policymakers with valuable information for understanding the key factors behind
heavy pollution, but it can also help modelers to identify the sources of uncertainties in numerical models.

## 1 Introduction

Atmospheric particulate matter, especially $PM_{2.5}$ (fine particulates less than 2.5 μm in diameter), can reduce visibility,
degrade air quality, threaten human health, and increase mortality (Xing et al., 2021; Huang et al., 2014; Lelieveld et al., 2015;
Evans et al., 2013; Fu et al., 2019; Janssen et al., 2013; Orellano et al., 2020). Over the past few decades, rapid industrialization
and urbanization have led to severe haze pollution in China (Lu et al., 2019b; Chen et al., 2018; Liu et al., 2017; Hartmann et
al., 2014). Beijing–Tianjin–Hebei (BTH) is one of the regions in China with the highest $PM_{2.5}$ concentrations (Lin et al., 2015;
Yang et al., 2020b). Annual concentrations of $PM_{2.5}$ in BTH reached 106 μg·m⁻³ in 2013, almost 3 times higher than China's
standard (35 μg·m⁻³) and 10 times higher than that of the World Health Organization (10 μg·m⁻³).
To mitigate the extremely severe and persistent haze in China and reduce air pollutant emissions, strict emission control
policies have been implemented by the Chinese government. However, the ambient $PM_{2.5}$ concentration is not only controlled
by emissions, but also largely influenced by chemical formation processes and unfavorable meteorological conditions
(Gelencsér et al., 2007; Jia et al., 2015; Wang et al., 2015; He et al., 2016; Sun et al., 2016). Numerous studies have stressed
the importance of chemical formation in the occurrence of severe haze events in China (Huang et al., 2014; Sun et al., 2016;
Chen et al., 2022). Unfavorable meteorological conditions associated with low wind speed, high humidity, temperature
inversion, and low planetary boundary layer can lead to weak atmospheric dispersion conditions and suppress the diffusion of
air pollutants (Chen et al., 2020b; Zheng et al., 2019). Moreover, the emissions, chemistry, and meteorological processes in
the atmosphere also interact with each other. For example, high humidity not only promotes hygroscopic growth but also gas-
to-particle partitioning, reflecting the correlation between the effect of physical and chemical processes on the concentration
of $PM_{2.5}$. These complex atmospheric processes demand that effective $PM_{2.5}$ control strategies must be formulated and adopted
on the basis of an in-depth understanding of the effects of meteorology, emissions, atmospheric chemistry, and their
interactions on the formation of $PM_{2.5}$. Although the basic relationships between $PM_{2.5}$ and different influencing factors have
been revealed, the quantitative influences of these factors on certain pollution episodes remains unclear, and it is difficult to
quantify and distinguish the roles of each factor because of their complex interactions and different behaviours from one case
to another (Li et al., 2011).



There have been some tools developed based on chemical transport models (CTMs) to analyse the effects of different factors
on PM$_{2.5}$ concentrations. The integrated process rate (IPR) method employed in the Community Multiscale Air Quality
(CMAQ) model can quantify the contributions of different physicochemical processes in numerical models, thus providing a
comprehensive understanding of the formation of air pollution (Jeffries and Tonnesen, 1994). The IPR method has been applied
to study the formation processes and mechanisms of O$_3$ and particulate matter in many cities (Liu et al., 2010; Li et al., 2014;
Fan et al., 2014; Huang et al., 2016; Chen et al., 2019a; Chen et al., 2019c; Fu et al., 2020). However, the IPR method can
obtain the contributions of different processes in a model, it ignores the nonlinear interactions between different processes,
which may lead to uncertain results.
From another perspective, the scenario analysis approach (SAA) has been employed to assess the response of PM$_{2.5}$ to
emission changes by changing the emission inventory of the model inputs under fixed meteorological fields, as well as the
response of PM$_{2.5}$ to meteorological changes by changing the meteorological fields under fixed emissions. For example, Zheng
et al. (2015b) found that the heavy pollution that occurred in winter 2013 in Northeast China was mainly caused by the stable
weather conditions in most parts of the region, rather than a sudden increase in anthropogenic emissions, through comparison
with the same period in 2012. Zhang et al. (2019a) reported that, although interannual meteorological changes may notably
affect the PM$_{2.5}$ concentration, the corresponding impact on the five-year trend of PM$_{2.5}$ concentration in China is relatively
limited (which they established by comparing results between the year 2017 and 2013). However, the traditional SAA method
is also incapable of analysing the nonlinear effects. Therefore, Stein and Alpert (1993) developed the Factor Separation (FS)
method to perform model sensitivity analysis and identify factors that contribute significantly to the model output. Compared
with the SAA method, the FS method is superior in dealing with nonlinear processes that involve two or more factors. By
performing multiple sensitivity experiments with different combinations of factors, the FS method allows one to assess the
impact of a single factor in a nonlinear system as well as the interaction between that factor and other factors. This method is
widely used in environmental and meteorological research (Romero et al., 2000; Alpert et al., 1999). For example, Tao et al.
(2005) assessed the amount of surface O$_3$ originated from area, mobile and point sources in the presence of biological emissions
and quantified the contributions of biogenic emissions and the synergy between anthropogenic and biogenic emissions (Tao
et al., 2003). The method can also be used to calculate the synergistic contributions of anthropogenic volatile organic
compounds, biogenic volatile organic compounds, and nitrogen oxides (NO$_x$) to surface O$_3$ (Qu et al., 2013). However, both
the SAA and FS method need to construct new simulation scenarios by changing the simulation conditions (emission source
or meteorological field) and uses the differences between the simulation results of different scenarios to represent the
contributions of the factors of interest. This means that the results of both the SAA and FS method are relative, being dependent
on the simulation scenario employed. For example, the meteorological conditions we used to construct the simulation scenarios
would alter the calculated contributions of meteorological processes to the PM$_{2.5}$ in the SAA and FS method. In addition, the
newly constructed simulation scenarios no longer represent the base simulation of the actual case because of the changed input
information of the CTM.





In addition to the methods that use CTMs, methods based on observations have also been developed. For example, the
PLMA (parameter linking air quality to meteorological conditions) index has been used to determine the contribution of
meteorology and emissions to air pollution (Zhang et al., 2015; Zhang et al., 2019b; Yang et al., 2016). Studies employing
principal component analysis or those targeting the correlation between $PM_{2.5}$ and meteorological elements have suggested
that a low wind speed and high humidity facilitate haze formation (Wang et al., 2013; Pang et al., 2009; Shu et al., 2017; Zhai
et al., 2019). Considering that a single meteorological element does not fully explain the relationship between meteorology
and $PM_{2.5}$, an artificial neural network model has been used to investigate the multiscale meteorological conditions, enabling
the meteorological influence to be quantified by the explained variance (He et al., 2017).
To date, none of the above methods can meet the following conditions at the same time: (1) on the premise of not changing
the base simulation conditions (without constructing other simulation scenarios as a reference system), we can quantitatively
analyze the contributions of meteorological, emission, and chemical processes to the variations of air pollutant concentrations
in an individual pollution case; (2) separation of the interactions between different factors; (3) the capability to analyse the
meteorological contribution given its considerable importance in analysing the pollution process; and (4) equality between the
sum of all analytical quantities and the simulated concentration change at any hourly time point so as to ensure that the
analytical quantity can fully reveal the reasons for the concentration change. In view of the different advantages and
disadvantages of these traditional methods mentioned above, we developed a novel quantitative decoupling analysis (QDA)
method and assessed the effects of emissions, meteorology, chemical reactions, and their interactions on the $PM_{2.5}$
concentration in a typical pollution case in Beijing. The QDA method tracks the change in $PM_{2.5}$ concentration in response to
changes in emissions, meteorological conditions, and chemical reactions in high-pollution cases. Thus, this method provides
a useful tool for identifying and quantifying the main determining factors of pollution cases, which can be used by decision-
makers for selecting the optimal scheme from different air pollution control and emergency response strategies. The differences
in QDA results of different model mechanisms can be compared to help identify the key process and improve its representation
in atmospheric models—for example, the physicochemical structure in the boundary layer and formation mechanism of
secondary air pollution (Chen et al., 2019a; Kang et al., 2019; Xing et al., 2017; Goncalves et al., 2009).
**2 Methods and data**
**2.1 Description of the QDA method**
In this section, we provide a detailed description of the QDA method proposed in this study, including its theoretical basis,
algorithms, and its realization in a model, as well as its relationship with the SAA, FS and IPR methods.
**2.1.1 Factors affecting $PM_{2.5}$ concentration and their contributions in CTMs**
The governing equation for CTMs is the three-dimensional semi-empirical Euler diffusion equation (Seinfeld and Pandis,
2016; Zhao et al., 2020):



$\quad \frac{\partial C_i}{\partial t} = -\left(u\frac{\partial C_i}{\partial x} + v\frac{\partial C_i}{\partial y} + w\frac{\partial C_i}{\partial z}\right) + \frac{\partial}{\partial x}\left(K_x\frac{\partial C_i}{\partial x}\right) + \frac{\partial}{\partial y}\left(K_y\frac{\partial C_i}{\partial y}\right) + \frac{\partial}{\partial z}\left(K_z\frac{\partial C_i}{\partial z}\right) + S + R_e - R_d - W_{ash}$ $\qquad$ …(1)
where $C_i$ is the concentration of species $i$ in the CTM; $u$, $v$ and $w$ are the wind velocity components in the $x$, $y$ and $z$ directions,
respectively; $K_x$, $K_y$ and $K_z$ are the diffusion coefficients in the $x$, $y$ and $z$ directions; $S$ denotes the direct emissions of $C_i$; $R_e$
is the chemical term, mainly affected by the chemical reaction mechanism; and $R_d$ and $W_{ash}$ are the dry and wet deposition
terms, respectively. Equation (1) is an instantaneous equation that cannot be solved analytically. In order to solve it numerically,
the differential equation is calculated by the finite-difference and operator splitting method in three-dimensional CTMs
(Santillana et al., 2016). We can define the advection operator as $\text{ADV} = -\left(u\frac{\partial}{\partial x} + v\frac{\partial}{\partial y} + w\frac{\partial}{\partial z}\right)$, the diffusion operator as
$\text{DIFF} = \frac{\partial}{\partial x}\left(K_x\frac{\partial}{\partial x}\right) + \frac{\partial}{\partial y}\left(K_y\frac{\partial}{\partial y}\right) + \frac{\partial}{\partial z}\left(K_z\frac{\partial}{\partial z}\right)$, the emission operator as EMIS, the chemical operator as CHEM, and the
deposition operator as DEP, and then Eq. (1) can be rewritten as:
$\quad \frac{\partial C_i}{\partial t} = \text{ADV}(C_i) + \text{DIFF}(C_i) + \text{EMIS}(C_i) + \text{CHEM}(C_i) + \text{DEP}(C_i)$ $\qquad$ …(2)
These model operators can also be classified in different ways. For example, the ADV, DIFF and DEP operators can be
combined and defined as the meteorological operator (MET), and then Eq. (2) would become:
$\quad \frac{\partial C_i}{\partial t} = \text{EMIS}(C_i) + \text{MET}(C_i) + \text{CHEM}(C_i)$ $\qquad$ …(3)
Furthermore, to produce refined process allocation, the DEP operator could be decomposed into a dry deposition operator and
wet deposition operator, and the CHEM operator could be decomposed into a gas-phase chemistry operator, liquid-phase
chemistry operator, or heterogeneous chemistry operator.
$\quad$ After the time is divided into model time steps, the calculation of Eq. (3) within one time step is carried out by the operator
splitting method in the order of EMIS, MET, and CHEM, as illustrated in Eqs. (4)–(6):
$\quad \frac{\partial C^1}{\partial t} = \text{EMIS}(C^1)$ $\qquad$ …(4)
$\quad \frac{\partial C^2}{\partial t} = \text{MET}(C^2)$ $\qquad$ …(5)
$\quad \frac{\partial C^3}{\partial t} = \text{CHEM}(C^3)$ $\qquad$ …(6)
where $C^1$ is the initial concentration of the specific species $i$ in a model step. If $\Delta C^1$ is the integration result of $\frac{\partial C^1}{\partial t}$ to $t$ in Eq.
(4) during the time step, then we can obtain $C^2 = C^1 + \Delta C^1$ and use it as the initial value to be input into the next operator,
MET, so the integration result of $\frac{\partial C^2}{\partial t}$, marked as $\Delta C^2$, would be affected by $C^1$ and $\Delta C^1$. Analogously, $C^3 = C^2 + \Delta C^2$. The
terms $\Delta C^1$, $\Delta C^2$ and $\Delta C^3$ correspond to the contribution of emissions, meteorology, and chemistry, respectively, by the IPR
method. However, if the calculation order between these operators is changed, different contribution results will be obtained.
This non-uniqueness of the contribution results comes from the nonlinear effects of different operator processes on the pollutant
concentration after the operator splitting. The concentration calculated by the latter operator process will be affected by the
results of the former operator processes. This nonlinear effect influenced by the former processes has not been separated in



previous research, which will bias the results regarding the contributions of different process operators (as with the IPR
method). To obtain more accurate and reliable results on the contributions of emissions, meteorological processes, and
chemical processes, it is necessary to quantify the nonlinear effects among these three operator processes.
If all three operators use the initial concentration of time step $C^1$ as the input value, just as in the following three equations:
$\frac{\partial C^1}{\partial t} = \text{EMIS}(C^1)$ …(7)
$\frac{\partial C^1}{\partial t} = \text{MET}(C^1)$ …(8)
$\frac{\partial C^1}{\partial t} = \text{CHEM}(C^1)$ …(9)
then the contribution results obtained by integration of Eqs. (7)–(9) only depend on the value of $C^1$ and are unaffected by the
nonlinear interactions of other operator processes. Therefore, the integration results of Eqs. (7)–(9) can be regarded as pure
contributions. The results obtained after the integration of Eqs. (7)–(9) are marked as $\Delta C^E$, $\Delta C^M$ and $\Delta C^C$, respectively, where
$\Delta C^E$ is the pure contribution of the emission process to the concentration, $\Delta C^M$ is the pure contribution of the meteorological
process to the concentration, and $\Delta C^C$ is the pure contribution of the chemical process to the concentration. On this basis, we
can further explore how to quantify the nonlinear interactions of different operator processes on the concentration, so as to
achieve a complete analysis of the amount of concentration change.
**2.1.2 Theoretical basis of the QDA method**
Considering the simulation of a haze case (base simulation), the simulated PM$_{2.5}$ concentrations at step $t$+1 (PM$_{2.5}^{t+1}$) can
be calculated by running all the processes in the model (including emissions, meteorology and chemistry) with the simulated
PM$_{2.5}$ concentration at step $t$ (PM$_{2.5}^t$) as the initial condition. Taking the calculation of one model step (from $t$ to $t$+1) as the
example, we can define the function $F$ to denote the simulated PM$_{2.5}$ concentration using PM$_{2.5}^t$ as the initial concentration,
where the information in parentheses represents the process operators that have been experienced in that time step:
$F(0,0,0) = \text{PM}_{2.5}^t$ (10)
$F(x_1, x_2, x_3) = \text{PM}_{2.5}^{t+1}$ (11)
Here, $F(x_1, x_2, x_3)$ represents the simulated PM$_{2.5}$ concentration obtained after the initial concentration has been subjected to
the processes of emission ($x_1$), meteorology ($x_2$), and chemistry ($x_3$) through this step; and $F(0,0,0)$ is equal to the initial
concentration because it has not been subjected to any process operator. Therefore, the variation in PM$_{2.5}$ concentration in the
base simulation can be written as:
$\Delta PM_{2.5}^{t+1} = PM_{2.5}^{t+1} - PM_{2.5}^t = F(x_1, x_2, x_3) - F(0,0,0)$ (12)
According to Taylor series expansion, the function $F$ can be decomposed as follows:

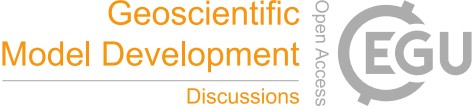

$F(x_1, x_2, x_3) - F(0,0,0) = \sum_{i=1}^{3} \frac{\partial F}{\partial x_i} x_i + \frac{1}{2!} \left( \sum_{i=1}^{3} \frac{\partial^2 F}{\partial x_i^2} x_i^2 + 2 \frac{\partial^2 F}{\partial x_1 \partial x_2} x_1 x_2 + 2 \frac{\partial^2 F}{\partial x_2 \partial x_3} x_2 x_3 + 2 \frac{\partial^2 F}{\partial x_1 \partial x_3} x_1 x_3 \right) +$
$\frac{1}{3!} \left( \sum_{i=1}^{3} \frac{\partial^3 F}{\partial x_i^3} x_i^3 + \sum_{a=1}^{2} 3 \frac{\partial^3 F}{\partial x_1^a \partial x_2^{3-a}} x_1^a x_2^{3-a} + \sum_{a=1}^{2} 3 \frac{\partial^3 F}{\partial x_2^a \partial x_3^{3-a}} x_2^a x_3^{3-a} + \sum_{a=1}^{2} 3 \frac{\partial^3 F}{\partial x_1^a \partial x_3^{3-a}} x_1^a x_3^{3-a} +$
$6 \frac{\partial^3 F}{\partial x_1 \partial x_2 \partial x_3} x_1 x_2 x_3 \right) + \cdots + o^n$                                   (13)
The terms in Eq. (13) that only contain partial derivatives to $x_1$ ($\frac{\partial F}{\partial x_1} x_1 + \frac{1}{2!} \frac{\partial^2 F}{\partial x_1^2} x_1^2 + \frac{1}{3!} \frac{\partial^3 F}{\partial x_1^3} x_1^3 + \cdots$) can be understood as
the pure emission contribution (marked as E); the terms that only contain partial derivatives to $x_2$ ($\frac{\partial F}{\partial x_2} x_2 + \frac{1}{2!} \frac{\partial^2 F}{\partial x_2^2} x_2^2 +$
$\frac{1}{3!} \frac{\partial^3 F}{\partial x_2^3} x_2^3 + \cdots$) can be understood as the pure meteorology contribution (marked as M); and the terms that only contain partial
derivatives to $x_3$ ($\frac{\partial F}{\partial x_3} x_3 + \frac{1}{2!} \frac{\partial^2 F}{\partial x_3^2} x_3^2 + \frac{1}{3!} \frac{\partial^3 F}{\partial x_3^3} x_3^3 + \cdots$) can be understood as the pure chemistry contribution (marked as C).
These pure contributions have the same meaning as $\Delta C^E$, $\Delta C^M$ and $\Delta C^C$ in the previous section, indicating the amount of
concentration change that occurs under the influence of only one process operator. The cross-derivative terms indicate the
effects of the interaction of different operators on the $PM_{2.5}$ concentration—for example, $\frac{1}{2!} \frac{2 \partial^2 F}{\partial x_1 \partial x_2} x_1 x_2 +$
$\frac{1}{3!} \sum_{a=1}^{2} \frac{3 \partial^3 F}{\partial x_1^a \partial x_2^{3-a}} x_1^a x_2^{3-a} + \cdots$ represents the interaction between emissions and meteorology on the concentration (ME), and
the term $\frac{1}{3!} \left( 6 \frac{\partial^3 F}{\partial x_1 \partial x_2 \partial x_3} x_1 x_2 x_3 \right) + \cdots$ represents the interaction among emissions, meteorology and chemistry on the
concentration (MCE). Detailed definitions of the terms in Eq. (13) are available in Table 1. Please note that "pure" in this
context means that, within a time step, the corresponding contribution is only due to the influence of a certain process operator
on the initial value and is unaffected by other operators. For example, the pure contribution of emissions (E) depends only on
local, direct emissions, and cannot represent the indirect contribution of emissions, which include the amount of $PM_{2.5}$
produced by the emitted precursor participating in the chemical reaction.
According to these definitions, the $PM_{2.5}$ variations from step $t$ to $t+1$ in the base simulation can be decomposed into
seven contributions, including the analytical quantities of $M$, $E$, $C$, ME, MC, CE, and MCE, as follows:
$\Delta PM_{2.5}^{t+1} = M + E + C + ME + MC + CE + MCE$                                   (14)
**2.1.3 Algorithms of the QDA method and its implementation in the model**
The QDA method uses algorithms similar to the FS method introduced by Stein and Alpert (1993) to calculate the
contributions in Eq. (14). By setting the parameters in the parentheses of $F$ to be $x_i$ ($i = 1,2,3$) or 0 to respectively represent
the concentration at time step $t+1$ with or without the corresponding process in the model, we can obtain the following
equations:
$F(x_1, 0,0) - F(0,0,0) = \frac{\partial F}{\partial x_1} x_1 + \frac{1}{2!} \frac{\partial^2 F}{\partial x_1^2} x_1^2 + \frac{1}{3!} \frac{\partial^3 F}{\partial x_1^3} x_1^3 + \cdots = E$                       (15)





$F(0, x_2, 0) - F(0,0,0) = \frac{\partial F}{\partial x_2} x_2 + \frac{1}{2!} \frac{\partial^2 F}{\partial x_2^2} x_2^2 + \frac{1}{3!} \frac{\partial^3 F}{\partial x_2^3} x_2^3 + \cdots = M$     (16)
$F(0, 0, x_3) - F(0,0,0) = \frac{\partial F}{\partial x_3} x_3 + \frac{1}{2!} \frac{\partial^2 F}{\partial x_3^2} x_3^2 + \frac{1}{3!} \frac{\partial^3 F}{\partial x_3^3} x_3^3 + \cdots = C$     (17)
$F(x_1, x_2, 0) - F(0,0,0) = \frac{\partial F}{\partial x_1} x_1 + \frac{\partial F}{\partial x_2} x_2 + \frac{1}{2!} \left( \frac{\partial^2 F}{\partial x_1^2} x_1^2 + \frac{\partial^2 F}{\partial x_2^2} x_2^2 + 2 \frac{\partial^2 F}{\partial x_1 \partial x_2} x_1 x_2 \right) + \cdots = E + M + \text{ME}$     (18)
$F(x_1, 0, x_3) - F(0,0,0) = \frac{\partial F}{\partial x_1} x_1 + \frac{\partial F}{\partial x_3} x_3 + \frac{1}{2!} \left( \frac{\partial^2 F}{\partial x_1^2} x_1^2 + \frac{\partial^2 F}{\partial x_3^2} x_3^2 + 2 \frac{\partial^2 F}{\partial x_1 \partial x_3} x_1 x_3 \right) + \cdots = E + C + \text{CE}$     (19)
$F(0, x_2, x_3) - F(0,0,0) = \frac{\partial F}{\partial x_2} x_2 + \frac{\partial F}{\partial x_3} x_3 + \frac{1}{2!} \left( \frac{\partial^2 F}{\partial x_2^2} x_2^2 + \frac{\partial^2 F}{\partial x_3^2} x_3^2 + 2 \frac{\partial^2 F}{\partial x_2 \partial x_3} x_2 x_3 \right) + \cdots = M + C + \text{MC}$     (20)
where $F(x_1, 0,0)$, $F(0, x_2, 0)$ and $F(0,0, x_3)$ can be calculated by the built-in scenario simulations that only consider emissions,
meteorology and chemistry from step *t* to *t*+1, respectively; and $F(x_1, x_2, 0)$, $F(x_1, 0, x_3)$ and $F(0, x_2, x_3)$ are calculated by
the built-in scenario simulation that does not include chemistry, meteorology or emissions from step *t* to *t*+1, respectively
(Table 2). The initial concentrations in the built-in scenario simulations will be updated by the values of the base simulation
at each time step, which ensures the resulting contributions are at the same concentration starting point and can be used to
analyse the hourly concentration change of the base simulation. The codes of the built-in scenario simulations are embedded
in the original code of the CTM and the initial concentration of the built-in scenario simulations at each time step can be
synchronously updated by the base simulation—something that cannot be done by the FS or other previous methods.
Therefore, the values of $F(x_1, 0,0)$, $F(0, x_2, 0)$, $F(0,0, x_3)$, $F(x_1, x_2, 0)$, $F(x_1, 0, x_3)$ and $F(0, x_2, x_3)$ can be obtained from
the results of the six built-in scenario simulations, and the values of $F(0,0,0)$ and $F(x_1, x_2, x_3)$ can be simply obtained from
the base simulation. Based on these equations above, the contributions of the four interactions in Eq. (14) can be calculated as
follows:
$\text{ME} = F(x_1, x_2, 0) - F(x_1, 0,0) - F(0, x_2, 0) + F(0,0,0)$     (21)
$\text{CE} = F(x_1, 0, x_3) - F(x_1, 0,0) - F(0,0, x_3) + F(0,0,0)$     (22)
$\text{MC} = F(0, x_2, x_3) - F(0, x_2, 0) - F(0,0, x_3) + F(0,0,0)$     (23)
$\text{MCE} = F(x_1, x_2, x_3) + \left( F(x_1, 0,0) + F(0, x_2, 0) + F(0,0, x_3) \right) - \left( F(x_1, x_2, 0) + F(x_1, 0, x_3) + F(0, x_2, x_3) \right) - F(0,0,0)$

233     (24)

The above formulae are all introduced based on one time step. The QDA method uses the above algorithm in each mode time
step, and outputs the contribution analysis results of the change in PM$_{2.5}$ concentration per hour. The initial concentrations of
not only PM$_{2.5}$ but also other species (all species contained in the CTM) in the built-in scenario simulations would all be
updated by base simulated values at the beginning of the new step. Finally, the QDA method's analytical results of the
variation at each step in the model output species, including PM$_{2.5}$, can be obtained. The relationships among the seven
contributions in Eq. (14) can also be shown visually (Fig. 1), in which the processes of emissions, meteorology and chemistry
are denoted by the three circles and the interactions among the different processes are denoted by the overlapping areas (Lunt
et al., 2021).





### 2.1.4 Relationship and differences with the SAA, FS and IPR methods

The similarity between the FS method and the QDA method is that they employ the same idea to separate the contributions of different processes, while the biggest difference between them is the target of the algorithm. The FS method commonly targets an "individual case", in which several sets of scenario simulations will run independently for several days or even longer with different input conditions for the factors of concern. The difference among these simulations due to the input conditions will gradually accumulate with the simulation time, and this cumulative amount is understood as the contribution of the condition difference for the entire individual case. The QDA method targets the "time step", in which the process operator is switched within the time steps of built-in scenario simulations and the concentration differences between the built-in scenario simulations of the same time step can reflect the process contribution but there is no transfer or accumulation of contribution between time steps. Therefore, the QDA method can not only obtain the process contribution for any given hour, but also the total contribution during the individual case.

The FS method has limitations in research and practical operations. Firstly, it can only study the relative contributions and not the absolute contributions. Relative contribution means the contribution expressed by the difference between two or more individual cases. Absolute contribution means the contribution of the process itself in an individual case. For example, by reducing or increasing specific emission sources, the concentration changes caused by the emission differences and their interactions could be obtained. If we want to study the influence of all emission sources in the geographical range of the model simulation settings, the FS method would have to construct a simulation scenario with a zero emission source, but this would lead to the concentrations of air pollutants only coming from the boundary and initial conditions in the CTM. So, after a period of simulation, the concentrations become extremely low, which is not what we want. To avoid this problem, the QDA method would synchronously update the initial concentration for the built-in scenario simulations by using the base simulation concentrations at each time step, which makes a certain process shut down for no more than one hour and ensures a physically meaningful result. To research the absolute contribution of meteorological conditions to air pollutant concentrations, we cannot construct a simulation scenario that completely closes the meteorological conditions through the FS method. FS can only be used to study the relative differences in concentration due to the changes in meteorological conditions. The QDA method has no such limitation; it is not only able to obtain the absolute contributions of operator processes at any time, but also able to calculate the relative differences in the contributions under different emission or meteorological scenarios. Secondly, FS can calculate the contribution for the case as a whole but cannot obtain the contribution for any specific hour in the case. The QDA method draws lessons from the idea of the IPR method in that it analyses the influence of factors in every time step and successfully solves the above problems.

By analysing the contribution of each process in the model, the IPR method can be used to resolve the contributions of different physical and chemical processes to the concentration change of every time step. Different from the fact that all physical and chemical processes in the real atmosphere are carried out almost simultaneously, the processes in CTMs are all carried out in sequence. The idea of the IPR method is that, in a time step, the operator processes are executed in sequence





according to the order in the model, and the concentration difference before and after the execution is calculated to represent
the contribution of a single process. This makes the IPR method unable to consider the effects of the nonlinear interactions
among different processes on pollutant concentrations. The order in which process operators are executed varies among
different CTMs. Assuming that in CTMs calculations are performed in the order of emission, meteorological and chemical
processes, the contribution of emissions obtained by the IPR method equals E in QDA, while the meteorological contribution
in the IPR (the concentration change caused by atmospheric advection, diffusion and deposition) equals M + ME in QDA, and
the chemical contribution equals C + CE + MC + MCE. Likewise, if one assumes that the CTM calculations are carried out in
the order of emission, chemical and meteorological processes, the contribution of emissions obtained by the IPR method equals
E in QDA, while the chemical contribution in the IPR equals C + CE in QDA, and the meteorological contribution equals M
+ MC + ME + MCE. The above two examples show that the IPR method cannot separate the interactions among different
processes, which leads to the interactions being included in the obtained IPR contributions.
To some extent, the QDA method could be seen as a combination of the FS method and IPR method. This method
combines the advantages of the IPR method for time-step analysis and the analytical advantages of FS for separating
interactions, but it is different from each of the two methods.
**2.2 Combination of the QDA and IPR methods**
The above QDA method can also be combined with the IPR method to resolve more detailed information. This is achieved
by applying the IPR method to each built-in scenario simulation. The premise is to ignore the nonlinear effect within one time
step. In Sect. 2.1, we showed that meteorological and chemical operators can be split into smaller sub-process operators—for
example, the meteorological process can be divided into advection, diffusion, dry and wet deposition processes; and the
chemical process can be divided into the gas- and aqueous-phase chemistry, thermodynamic equilibrium processes, and
secondary organic aerosol (SOA) reactions. That is to say, the IPR analysis can be used in the operators of emissions,
meteorology and chemistry under the calculation framework of the QDA method at the same time (Fig. 2). Therefore, we can
obtain the sub-process contributions among the seven quantitative analytical factors in Eq. (14).
The results of the base simulation and each built-in scenario simulation at $t+1$ can be decomposed by IPR as follows:
$F(x_1, 0, 0) - F(0,0,0) = \text{emit}_{S_1}$                        (25)
$F(0, x_2, 0) - F(0,0,0) = \text{advhor}_{S_2} + \text{advvert}_{S_2} + \text{difhor}_{S_2} + \text{difvert}_{S_2} + \text{wetdep}_{S_2} + \text{drydep}_{S_2}$  (26)
$F(0, 0, x_3) - F(0,0,0) = \text{gaschem}_{S_3} + \text{ISORR}_{S_3} + \text{SOA}_{S_3}$             (27)
$F(x_1, x_2, 0) - F(0,0,0) = \text{emit}_{S_{12}} + \text{advhor}_{S_{12}} + \text{advvert}_{S_{12}} + \text{difhor}_{S_{12}} + \text{difvert}_{S_{12}} + \text{wetdep}_{S_{12}} + \text{drydep}_{S_{12}}$ (28)
$F(x_1, 0, x_3) - F(0,0,0) = \text{emit}_{S_{13}} + \text{gaschem}_{S_{13}} + \text{ISORR}_{S_{13}} + \text{SOA}_{S_{13}}$       (29)
$F(0, x_2, x_3) - F(0,0,0) = \text{advhor}_{S_{23}} + \text{advvert}_{S_{23}} + \text{difhor}_{S_{23}} + \text{difvert}_{S_{23}} + \text{wetdep}_{S_{23}} + \text{drydep}_{S_{23}} + \text{gaschem}_{S_{23}} +$
$\text{ISORR}_{S_{23}} + \text{SOA}_{S_{23}}$                         (30)





$F(x_1, x_2, x_3) - F(0,0,0) = \text{emit}_S + \text{advhor}_S + \text{advvert}_S + \text{difhor}_S + \text{difvert}_S + \text{wetdep}_S + \text{drydep}_S + \text{gaschem}_S +$
$\text{ISORR}_S + \text{SOA}_S$ (31)
Table 2 explains the meaning of each item on the left-hand sides of Eqs. (25)–(30); $F(x_1, x_2, x_3)$ and $F(0,0,0)$ represent the
PM$_{2.5}$ concentration at time $t$+1 and time $t$ in the base simulation S; the subscripts on the right-hand sides of Eqs. (25)–(30)
denote the corresponding simulation mark; the IPR terms refer to previous research (Chen et al., 2019a; Chen et al., 2019c);
and these subprocess definitions and abbreviations are detailed in Table 3. Combining Eqs. (25)–(31) and Eqs. (15)–(25), the
contributions of sub-process operators in any QDA analytic quantity can be obtained.
**2.3 Model setup and emission inventories**
To illustrate the use of the QDA method, we embedded its codes into the Nested Air Quality Prediction Modeling System
(NAQPMS) model and built QDA v1.0 for NAQPMS. The QDA method can be combined with other CTMs in a similar way
following the QDA algorithm. NAQPMS is a three-dimensional regional Eulerian CTM developed by the Institute of
Atmospheric Physics, Chinese Academy of Sciences, which has been widely used in scientific research and operational air
quality prediction (Wang et al., 2014; Du et al., 2021; Kong et al., 2021; Wang et al., 2021; Akimoto et al., 2020; Yang et al.,
2020a) owing to its good performance in simulating the emission, meteorological and chemical processes in the atmosphere.
Within the model, the gas-phase chemistry is simulated by the "carbon bonding mechanism Z" developed by Zaveri and Peters
(1999), which includes 134 reactions and 71 species. For inorganic aerosols, the ISORROPIA v1.7 thermodynamic equilibrium
module (Nenes et al., 1998) is used to simulate the ammonia–nitrate–sulfate–chloride–sodium–water system. Six SOAs are
processed by a two-product module in NAQPMS (Odum et al., 1997). The aqueous-phase chemistry and wet deposition are
modelled using the Regional Acid Deposition Model mechanism in CMAQ version 4.6. The dry deposition of gases and
aerosols is simulated based on the scheme of Wesely (1989) and the advection is simulated with an accurate mass-conservation
algorithm from Walcek and Aleksic (1998). More technical details on NAQPMS could be found in Li et al. (2012).
To illustrate the feasibility of the QDA method and quantitatively analyse the magnitudes of the contributions from
emissions, meteorology and chemistry to the variation in PM$_{2.5}$ during heavy pollution, we applied the method to a week-long
heavy-haze episode that took place in Beijing during 17–28 February 2014. Figure 3 shows the modelling domain of this case,
which covers most of East Asia with a horizontal resolution of 45 km. Vertically, NAQPMS uses 20 nonequally distributed
terrain-following layers from the surface (~100 m) to 20 km. The anthropogenic emission inventories used in the simulation
were obtained from the Chinese Multi-resolution Emission Inventory (MEIC) for the year 2014 developed by Tsinghua
University (http://www.meicmodel.org). We adjusted the original inventory with reference to the diurnal profile of the
emission inventory in MICS-Asia III (Model Inter-Comparison Study for Asia III), which is shown in Fig. S1. Biogenic
emissions were obtained from the Model of Natural Gas and Aerosol Emissions (MEGAN v2.0) (Guenther et al., 2006), and
the biomass burning emissions were obtained from the the Global Fire Emissions Database version 4 (Randerson et al., 2017;
van der Werf et al., 2010). A clean initial condition was used in the simulation with a 10-day free run of NAQPMS as a spin-
up time. The top and boundary conditions of the outermost region were extracted from the global CTM MOZART (Model for





Ozone and Related Chemical Tracers) version 2.5, with a 3-h temporal resolution (Brasseur et al., 1998). The offline hourly
meteorological fields were generated by the Weather Research and Forecasting (WRF) model version 3.7 (http://www.wrf-
model.org/), driven by National Centers for Environmental Prediction (NCEP) Final Analysis data (FNL).

**2.4 Observation data**

The observational data used in this study included surface observations of $PM_{2.5}$, particulate matter smaller than 10 μm
in diameter ($PM_{10}$), $NO_2$, $O_3$, $SO_2$ and CO obtained from the China National Environmental Monitoring Center. Surface
observations of wind speed, wind direction, temperature, relative humidity, and station pressure; and vertical observations of
wind speed, wind direction, temperature, and relative humidity, were retrieved from the China Meteorological Administration.
The spatial distributions of the meteorological and air quality observation sites are shown in Fig. 3. To compare with the $PM_{2.5}$
observations, the simulated $PM_{2.5}$ concentrations were comprised of primary $PM_{2.5}$ (including black carbon, primary organic
aerosol, and other directly emitted $PM_{2.5}$) and secondary $PM_{2.5}$, including sulfate, nitrate, ammonium, and SOA produced by
chemical reactions.

**3 Results and discussion**

**3.1 Observed pollution during the heavy-haze episode**

A serious pollution event occurred in the Beijing area during 19–27 February 2014, with the observed mean $PM_{2.5}$
concentration reaching 168.9 μg m$^{-3}$, more than double the national secondary standard level (75 μg m$^{-3}$). As shown in Fig.
S2, this pollution episode also affected a wide area of the BTH region, with severe haze mostly located in the southern part of
the region before 23 February and gradually extending northwards to encompass wider areas. The $SO_2$ and $NO_2$ concentrations
did not exhibit notable exceedances as the $PM_{2.5}$ did, indicating that this case was a typical particulate-led pollution event.
To investigate the characteristics of the contributions from meteorology, emissions and chemistry in different stages of
this haze event, we divided the whole episode into four stages based on the temporal characteristics of the $PM_{2.5}$ concentration
in Beijing (Fig. 4): (1) the pre-contamination stage [0800 LST (local standard time) 17 February to 1400 LST 19 February]
with relatively low $PM_{2.5}$ concentrations and flat variation; (2) the accumulation stage (1500 LST 19 February to 0800 LST 23
February) when the $PM_{2.5}$ concentration increased the most rapidly; (3) the pollution maintenance stage (0900 LST 23 February
to 1800 LST 26 February) when the $PM_{2.5}$ concentration remained high with small fluctuations; and (4) the pollution removal
stage (1900 LST 26 February to 0800 LST 27 February) when the $PM_{2.5}$ concentration rapidly dropped.

**3.2 Validation of the meteorology and chemistry simulations**

To assess the accuracy of the model, simulated meteorological parameters and air pollutant concentrations were compared
with observed values. We used several evaluation indicators to quantitatively assess the model performance, including the
simulated mean, observed mean, correlation coefficient ($R$), mean fractional bias (MFB), mean fractional error (MFE), mean



bias, mean error (MEr), normalized mean bias (NMB), normalized mean error (NME), root-mean-square error, and index of
agreement (IOA), which are defined in Table S1. The verification results of meteorological elements are shown in Table S2,
revealing the $R$ of temperature (Temp), relative humidity (RH) and pressure to all be above 0.85. The correlation between wind
speed (WS) and observation data ($R$=0.47) is better than that of wind direction (WD: $R$=0.24). Although the MEr of the
simulated wind is greater than that of other meteorological elements, the NME and NMB are less than 1, which indicates that
the simulation and observation match well on the whole, and the MEr may have little influence on the performance of aerosol
simulation.
The simulations based on the NAQPMS model generally reproduced the magnitude of, and temporal variation in, the
$PM_{2.5}$ concentration in the Beijing area well, with an $R$ of approximately 0.83. The model simulation results exhibit relatively
larger underestimations of the $PM_{2.5}$ concentration from 20–23 February, which may be attributable to the overestimation of
the simulated wind speed by the WRF model during this period (Figs. S3 and S4). Regarding the two important precursors of
$PM_{2.5}$, the simulated $NO_2$ and $SO_2$ concentrations also agree well with the observations, with $R$ values of approximately 0.71
and 0.76, respectively. In general, the simulated $PM_{2.5}$ concentrations satisfy the NMB, NME, $R$, and IOA performance
standards (NMB<20%, NME<45%, $R$>0.6, and IOA>0.7) proposed by Huang et al. (2021a), and the simulated $SO_2$ and $NO_2$
concentrations all satisfy the MFB and MFE performance standards (MFB<30%, MFE<50%) proposed by (Boylan and Russell,
2006). The simulated sulfate, nitrate and ammonium concentrations were also compared with observations, to evaluate the
chemical processes in the NAQPMS model (Fig. S5). The model reproduced the variation in secondary inorganic aerosols
(SIAs) well during this episode ($R$>0.82), although the model underestimated the sulfate concentration, possibly due to missing
reaction pathways of sulfuric acid in the model, such as heterogeneous chemistry (Zheng et al., 2015a; Cheng et al., 2016).
Underestimation of the sulfate concentration is a common problem in current CTMs (Chen et al., 2019b), but one that is beyond
the scope of this study. However, this could lead to uncertainty in the estimation of the contribution from chemistry to the
$PM_{2.5}$ concentration. In summary, the simulation suitably reproduced the evolution of this pollution process from the pre-
contamination period to the accumulation, maintenance, and removal periods, which laid a good foundation for subsequent
analysis of the physical and chemical processes.

### 393 3.3 Temporal variation of the QDA results in different stages

Figure 5 shows the time series of the calculated contributions from emissions, meteorology, chemistry, and their
interactions to the hourly variation in $PM_{2.5}$ concentration using the QDA method. We can clearly see that in Fig. 5(b) the sum
of all contributions is exactly equal to the hourly change in the $PM_{2.5}$ concentration, indicating that the QDA method can fully
resolve the variation in the $PM_{2.5}$ concentration.
The characteristics of temporal variation vary among different factors. Among the seven QDA analytical factors, the
fluctuation range of M is the largest, which ranges from −48.7 to 7.4 $\mu g \cdot m^{-3} \cdot h^{-1}$. When the change in $PM_{2.5}$ concentration is
positive, M plays a role in promoting the accumulation of the $PM_{2.5}$ concentration. When the change in $PM_{2.5}$ is negative, M
must play a clearing role. On the hourly scale, the value of E is not always the biggest. For example, on the afternoons of 23–



25 February, CE is significantly larger than E. However, on average, E is the largest mean contribution with the highest value
of 0.88 μg·m$^{-3}$·h$^{-1}$ in two stages (Table 4). The contribution from chemistry exhibits remarkable diurnal variation, being
notably larger during the daytime than at nighttime. This occurs because the atmospheric oxidation capacity during daytime is
higher than at nighttime, which is more conducive to secondary PM$_{2.5}$ formation (Huang et al., 2021b; Chen et al., 2020a; Lu
et al., 2019a), and similar conclusions have been reported in other modelling studies (Chen et al., 2019a; Li et al., 2014).
We investigated the influences of the nonlinear effects on the PM$_{2.5}$ concentration by summing all the contributions of
the interactions among the different physical and chemical processes (COUP=EM+CE+MC+MCE). Figure 6 and Table 5
show the QDA results in the different stages of the episode. The M and EM exhibit a notable negative contribution to PM$_{2.5}$ in
the first stage, which was enough to remove the newly emitted or formed PM$_{2.5}$ from emissions and chemical reactions
(|M+EM|>|E+C+CE+MCE| in Table 4). Thus, the PM$_{2.5}$ concentration was relatively low in the first stage. However, M shifts
to a positive contribution in stage 2, and there are no other removal processes except EM during this stage. The average increase
in PM$_{2.5}$ per hour (M+E+C+CE+MC+MCE=1.87 μg·m$^{-3}$·h$^{-1}$) is significantly greater than the removal speed (EM = −0.08
μg·m$^{-3}$·h$^{-1}$), which led to rapid accumulation of the PM$_{2.5}$ concentration. Then, M becomes negative and acts as a cleaner in
stage 3, which nearly offsets the increase caused by E+C+COUP with a speed about 1.85 μg·m$^{-3}$·h$^{-1}$. Hence, the PM$_{2.5}$
concentration remained at a relatively steady level. In stage 4, the removal effects of M are much larger than those in the
previous stages owing to the strong northwesterly nonpolluted wind, leading to a rapid decline in the PM$_{2.5}$ concentration.
According to the IPR results (Figs. 6e–h), horizontal advection was the main removal process in M during stages 1 and 4,
indicating that horizontal outward transportation facilitated PM$_{2.5}$ reduction during the relative cleaning period (Chen et al.,
2020c). Vertical advection was the main accumulation process in M during stages 1, 2 and 4, while in stage 3 it had removal
effects on PM$_{2.5}$. It has been reported by Platis et al. (2016) that the downward transport of particles may be an important
reason for increased PM$_{2.5}$ in the entire atmospheric boundary layer. Figure 7(a) shows that, during stage 2, PM$_{2.5}$ is transported
from outside to the *in situ* area below the height of L7 via horizontal advection, while it is exported from outside in the upper
layer. These positive and negative results cancel each other out and make the horizontal advection contribute little to the entire
layer during this stage. PM$_{2.5}$ can originate from other places via long-distance transport (Du et al., 2020), which would lead
to weakening of boundary layer turbulence and thereby the facilitation of local pollution accumulation (Huang et al., 2020). It
can also be seen that the growth rate of PM$_{2.5}$ in L7–L9 is the highest, which is consistent with previous findings that the
accumulation of aerosols near the top of the boundary layer has the largest rate of increase (Liu et al., 2020). In addition, the
hourly contribution of wet deposition was zero and played a negligible role in the variation in different stages, which was due
to there being no precipitation during this typical severe haze event. The C yielded positive contributions in the first three
stages (0.23–0.37 μg·m$^{-3}$·h$^{-1}$) owing to the generation of SIAs and SOAs. In stage 4, C became negative (−0.08 μg·m$^{-3}$·h$^{-1}$) as
the environmental conditions at this time were suitable for nitrate decomposition (Chen et al., 2020c). According to their
definitions, C reflects the contribution from chemistry to PM$_{2.5}$ made by pre-existing gases in the atmosphere, and CE reflects
the same but made by newly emitted gases. Therefore, the larger the ratio of CE/C, the more efficient the chemical conversion.
The results suggest that the conversion efficiency of secondary aerosols was highest during stage 3 under the most serious





pollution, which is consistent with the results of other studies of heavy haze (Huang et al., 2014; Zhou et al., 2022; Shang et al., 2021).

It is worth noting that E only contains the contributions from direct emissions of $PM_{2.5}$ in local space, which was determined by the emissions inventory in the model. This definition is different from the contributions of emissions in previous studies, which also included the nonlinear effects between direct emissions and other processes (Maji et al., 2020; Zhang et al., 2018). From the perspective of tracing back to the sources, the ultimate source of pollutants is only the emissions, but the directly emitted $PM_{2.5}$ or precursors can affect other areas through meteorological transmission or chemical reactions. From a process viewpoint, it is obviously not just the emission process that should be involved. The hourly mean contribution of CE was largest during stage 3, and thus the implementation of emissions reduction during that stage would have greater weakening effects on the chemical generation of $PM_{2.5}$ than in the previous two stages. M reflects the net changes in $PM_{2.5}$ concentration resulting from pollutants following air masses in and out of the grid boxes of the model; plus, it is also the main way to reduce air pollution most of the time. Although its specific application in developing emission reduction strategies is not the focus of this paper, it is nonetheless worth highlighting that it can provide valuable insights into these related issues.

**3.4 Decoupling of nonlinear effects at different stages**

There are also nonnegligible nonlinear effects in each stage, and their contributions can sometimes even exceed pure contributions of meteorology, emissions, and chemistry. On the one hand, these nonlinear interactions are determined by the calculation method; whilst on the other hand, they are physically explainable. When emissions increase the concentrations of pollutants in the atmosphere, the amounts of pollutants transported by air masses will also increase, which is reflected by the nonlinear effect of EM. The emission process may increase the concentrations of precursors in the atmosphere. Based on the IPR results, CE reflects that newly emitted precursors produce secondary aerosols through chemical reactions and equilibrium partitioning. MC consists of two parts: the first part is the influence of meteorology on chemistry, in which meteorological processes can increase chemical production by transporting more precursors or decrease chemical production by reducing the concentrations of local precursors; while the second part involves the influence of chemistry on meteorology, since chemical processes can lead to an increase in the concentrations of secondary aerosols in the atmosphere. This may lead to an increase in pollutants carried by air masses in the corresponding region. MCE includes all meteorological, emission and chemical process interactions, which are complex and yield very small contributions. The hourly value of COUP ranged from −1.83 to 2.44 $\mu g \cdot m^{-3} \cdot h^{-1}$ during this haze episode, with an average value of approximately 0.30 $\mu g\ m^{-3}\ h^{-1}$. The nonlinear effect was shown to increase continuously from the beginning to heavy polluted periods. According to Table 4, from stage 1 to 3, the hourly mean value of COUP increased from 0.05 to 0.74 $\mu g \cdot m^{-3} \cdot h^{-1}$, and its proportion in the hourly variation of $PM_{2.5}$ also increased (from −3.68% to 740%).

During the entire episode, CE exhibited the largest nonlinear effect (0.27 $\mu g \cdot m^{-3} \cdot h^{-1}$ on average) and increased with the concentration of $PM_{2.5}$, indicating that the interactions between emissions and chemistry play an important role during heavy haze. According to the vertical distribution of CE in stage 2 (Fig. 7), the contribution of CE decreased from the surface to the





upper levels owing to the vertical reductions in air temperature and emissions. The contribution of MC revealed the largest
variation, with a fluctuational range up to 4.24 µg·m$^{-3}$·h$^{-1}$, because both the meteorology and chemistry are greatly influenced
by diurnal variation. As shown in Fig. 7, MC also indicated that meteorological processes could decrease the chemical process
in the surface layer and strengthen chemical formations in the upper layers (L3–L8), which could also be related to the
phenomenon in Fig. 7(a) that meteorological processes transport PM$_{2.5}$ and precursors from the lower layer to the upper layer.
EM suggests that local emissions may enhance the vertical diffusion of PM$_{2.5}$ from the surface layer to the upper layer. Primary
emitted PM$_{2.5}$ mainly occurred in the near-surface layers where the vertical wind speed was so low that vertical advection was
extremely limited. Thus, PM$_{2.5}$ emitted in the near-surface layers could reach the upper layers only through the process of
vertical diffusion.
In previous studies, the investigation of nonlinear effects was usually ignored when analysing heavy haze. The present
QDA results demonstrate that ignoring these nonlinear effects may cause bias when studying the pure contributions of
meteorology, emissions or chemistry to PM$_{2.5}$. For example, when discussing the effect of the pure contribution of emissions
on PM$_{2.5}$, if the effects in CE, EM and MCE are ignored, an uncertainty ranging from −0.86 to 1.86 µg·m$^{-3}$·h$^{-1}$ (CE +EM+MCE)
occurs on the hourly scale, especially during the worst polluted period, and this uncertainty may accumulate with time. This
suggests that quantitative analysis of the nonlinear effects is necessary to evaluate the process contributions in heavy-haze
episodes.

### 3.5 Discussion and evaluation of QDA

#### 3.5.1 Chemical compositions

The E calculated by the QDA method is influenced directly by the emissions inventory used in the simulations. Thus, we
mainly evaluated the calculated contributions of M and C in this study. However, there were no observational data linked to
the pure contributions of emissions or chemistry that could be used to verify the QDA method directly. Hence, the method was
evaluated with indirect results. Since the contribution from chemistry to PM$_{2.5}$ is mainly related to the formation of secondary
aerosols, the conversion rates of nitrate (NOR) and sulfate (SOR), as defined in Eqs. (32) and (33), were calculated to evaluate
the temporal variation in the chemical contribution obtained with the QDA method. Daily PM$_{2.5}$ composition data measured
by the Beijing Ecological Environment Monitoring Center were used to calculate NOR and SOR values in the different stages
of this haze episode:
$\text{NOR} = \frac{\text{NO}_3^-}{\text{NO}_3^- + \text{NO}_2}$                                                          (32)
$\text{SOR} = \frac{\text{SO}_4^{2-}}{\text{SO}_4^{2-} + \text{SO}_2}$                                                          (33)
We found that NOR and SOR increased by 0.09 and 0.02, respectively, from stage 1 to stage 2. NOR and SOR both
reached their maximum value (0.54 and 0.38, respectively) in stage 3. In stage 4, NOR and SOR both experienced a significant
decline. Other haze cases have also revealed that SOR and NOR greatly increased with PM$_{2.5}$ concentration (Song et al., 2019;





Xu et al., 2017; Yan et al., 2015a) and the proportion of secondary aerosols often increases with worsening haze (Xu et al.,
2019a; Li et al., 2017). Process analysis has also shown that the chemical reactions of $PM_{2.5}$ in the WRF-Chem model are
stronger during the day than that at night (Chen et al., 2019a), which is consistent with this study. In the QDA results, the
amount that can represent the chemical reaction intensity is C+CE. It can be seen that its total contribution had been increasing
from stage 1 to stage 3, and in stage 4 had decreased to its lowest level. This evidence, together with the QDA analysis results,
explains the importance of chemical reactions in heavy haze (Huang et al., 2019).

We also analysed the QDA results for SIAs, including nitrate, sulfate and ammonium, as well as their precursors,

including $NO_x$, $SO_2$, and $NH_3$, to provide further insight into the roles of chemical formations during haze episodes. Figure 8
shows the QDA results for SIAs, as well as their precursors, during the different stages of the episode. Notably, there were
small contributions of E to the sulfate concentrations because we parameterized 2.5% of sulfate primary emissions to consider
the particle formation on the sub-grid scale. As we can clearly see from Fig. 8, the chemical production of nitrate, sulfate and
ammonium agreed well with the chemical depletion of their precursors, suggesting good capability of the QDA method in
representing the chemical processes in the model. For example, during the first stage, the values of C for $NO_x$, $SO_2$, and $NH_3$
were all negative where the C values for nitrate, sulfate and ammonium were positive, reflecting the conversion of reactive
gases to $PM_{2.5}$. Consistent with the QDA results for $PM_{2.5}$ concentration, the QDA results for SIAs and their precursors showed
that chemistry provided an increasingly important role in the elevation of $PM_{2.5}$ concentrations. From stage 1 to stage 2, the
values of C for $NO_x$ and $SO_2$ changed from −0.18 to −0.27 μg·m$^{-3}$·h$^{-1}$ and from −0.01 to −0.02 μg·m$^{-3}$·h$^{-1}$, respectively.
Correspondingly, the values of C for nitrate and sulfate increased from 0.21 to 0.26 μg·m$^{-3}$·h$^{-1}$ and from 0.02 to 0.03 μg·m$^{-3}$·h$^{-1}$,
respectively. Consistent with the NOR and SOR analysis, chemical processes yielded the largest contribution during stage 3,
in which the values of C for $NO_x$ and $SO_2$ reached −0.45 and −0.06 μg·m$^{-3}$·h$^{-1}$, respectively, which was 66.7% and more than
twice as much as during stage 2. Correspondingly, the C value for sulfate increased from 0.03 to 0.08 μg·m$^{-3}$·h$^{-1}$ from stage 2
to stage 3. However, the C value for nitrate and ammonium was found to decrease in stage 3. In addition, the values of CE for
nitrate and ammonium were much larger in stage 3 than during stage 1 or stage 2, which reached up to 0.46 and 0.15 μg·m$^{-3}$·h$^{-1}$,
respectively. More $NH_3$ was also consumed by the interaction between chemistry and emissions during stage 3, with the value
of CE reaching −0.15 μg·m$^{-3}$·h$^{-1}$. This is because $NH_3$ was deficient during stage 3. Although more $NO_x$ was oxidized to $HNO_3$
during stage 3, most of the newly formed $HNO_3$ remained in the gas phase owing to the limited $NH_3$, leading to small C value
for nitrate but large C values for $NO_x$. In addition, the newly emitted $NH_3$ would react quickly with the pre-existing $HNO_3$ to
form nitrate and ammonium. That is why the values of CE for nitrate and ammonium were much larger in stage 3 than in
previous stages. On the contrary, stage 1 and stage 2 were in an $NH_3$-rich condition, so the newly formed $HNO_3$ and $H_2SO_4$
could react with the sufficient pre-existing $NH_3$ to form nitrate and sulfate without relying on fresh emissions of $NH_3$. Therefore,
there was good consistency between the C values of precursors and SIAs during stage 1 and stage 2. These results suggest that
the QDA method is capable of reflecting different chemical environments during different stages of haze episodes, and
emphasize that different emission control strategies should be adopted in different stages. For example, strict emissions control
should be performed for $NO_x$ and $SO_2$ emissions during stage 1 and stage 2, while during stage 3, when the $PM_{2.5}$ concentration





is highest, the control of NH$_3$ emissions would be a more efficient approach. The high efficiency of reducing NH$_3$ emissions
in alleviating heavy haze has been attested in studies based on both observations and model results (Liu et al., 2022; Xu et al.,
2019b; Qi et al., 2023; Zhai et al., 2021; Ge et al., 2019). However, these studies did not elucidate when is the most effective
time to control NH$_3$. Not only can the QDA method quantitatively explain the role of NH$_3$ in heavy haze, but it can also provide
valuable information on when and where controlling NH$_3$ emissions is more effective. Therefore, this method can provide
policymakers with valuable insights into the development of efficient emission control strategies during different stages of
pollution.
**3.5.2 Meteorological processes**
The contributions of meteorological processes were quantitively evaluated via the analysis of weather conditions. Figure
S6 clearly shows that, during stage 1, Beijing and its surrounding areas were influenced by a high-pressure system in
northeastern Inner Mongolia and a low-pressure system in the southwest with high wind speeds, which promoted PM$_{2.5}$
advection across the Beijing area. With the low-pressure system in Inner Mongolia moving slowly eastwards and finally
disappearing under the influence of westerly winds, Beijing was increasingly controlled by a uniform pressure field and
affected by weak southerly winds, which facilitated the transportation of air pollution from the southern BTH region to Beijing.
The small-scale high-pressure centre to the north of Beijing also blocked the airflow originating from the south, leading to the
accumulation of air pollutants in Beijing, which is consistent with the positive pure meteorological contribution (M>0) in stage
2. The potential source contribution function (PSCF) index can reflect the potential contribution of the inflow trajectory,
revealing that Baoding, Shijiazhuang and Cangzhou in Hebei in southern Beijing were the main sources of PM$_{2.5}$ transmission
in this case (Yan et al., 2015b). Research revealed that the transportation process in this case under the influence of weak
southerly winds from 19 to 20 February, along with the Parameter Linking Air-quality to Meteorological conditions/haze index
(PLAM), indicated a positive correlation between PM$_{2.5}$ and atmospheric stability (Zhong et al., 2018b). An inversion layer
occurred owing to the radiative cooling effect of the transported particles, which further aggravated aerosol accumulation
(Zhong et al., 2018a) (Fig. 9). The key role of transmission in the formation of high concentrations of PM$_{2.5}$ has also been
found in other haze events  (Sun et al., 2016; Huang et al., 2020; Zhang et al., 2019b).
In stage 3, the northern high-pressure system was compressed by the northwest low-pressure air system and moved to the
southeast sea area. The isobaric lines in Beijing became increasingly dense and the wind speed increased, which was conducive
to the diffusion of pollutants (M<0). However, due to the positive contribution of emissions and chemistry, the air quality did
not improve significantly. In stage 4, the northeast low-pressure system continued to develop and intensified, confronting the
Mongolian high-pressure system, resulting in a strong northwesterly airflow in North China that transported air pollutants to
the southeast sea area and greatly improved the air quality in Beijing. Therefore, the hourly contribution of M at this stage was
the largest, reflecting a strong cleaning effect. This is also consistent with the analysis of this pollution case in other studies
(Zhong et al., 2018b; Zhong et al., 2018a).





## 4 Conclusions and perspectives

In this study, a new QDA method targeting $PM_{2.5}$ was developed and applied to analyse a typical heavy-pollution case in Beijing. By quantitatively decomposing the pure contribution of meteorology, chemical reactions, emissions, and their nonlinear interactions in the hourly change of the $PM_{2.5}$ concentration, the formation process of heavy haze can be analysed from a new perspective. The QDA method innovatively combines the advantages of the FS and IPR methods and highlights the differences and connections between pure contributions and nonlinear interactions in air pollution problems from the perspective of process contributions and conservation of mass as a constraint.

The atmosphere is a typical nonlinear system. Unfavorable meteorological conditions are a frequently discussed issue in haze events and their quantification can be biased by nonlinear effects such as EM and MC. Separating pure contributions and nonlinear interactions can clearly reveal the timing and effect of favorable or unfavorable meteorological conditions. We divided the haze event in this study into four stages according to the characteristics of $PM_{2.5}$ concentration. It was found that the M during the accumulation stage (stage 2) was 0.34 $\mu g \cdot m^{-3} \cdot h^{-1}$, while in other stages it was negative on average, indicating that the pure meteorological contribution only in the accumulation stage favored the accumulation of $PM_{2.5}$. This means M mainly acts as a cleaner for $PM_{2.5}$ most of the time. However, when M continues for a period of time without removing pollution (M>0), $PM_{2.5}$ would lose its main mechanism to descend and therefore tend to grow rapidly under the superimposed influence of emissions and chemical processes, which would probably become the beginning of a heavy pollution event. Commonly, the effect of meteorological accumulation is the direct cause of haze formed by transportation and accumulation of $PM_{2.5}$, and QDA provides a clearer interpretation of this. For the atmosphere of the entire boundary layer in particular, the direct cumulative effect of M on $PM_{2.5}$ is not high. M usually plays the role of the most efficient cleaner, but it is no longer effective under the circumstances of unfavorable meteorological conditions, resulting in the $PM_{2.5}$ (which formed by emissions and chemical reactions) not being cleaned up in time, which is why unfavorable meteorological conditions may play a dominant role in the formation of haze. The aim of this study was to develop a new analysis method rather than study its application, so QDA was only applied to one typical haze event, meaning more cases in different regions and periods should be studied in the future. The consideration of nonlinear effects provides a useful way to handle the nonlinear characteristics of the atmosphere, thus filling the gaps in traditional methods in terms of nonlinear uncertainty. The importance of nonlinear effects includes, firstly, eliminating the interference of other processes in quantifying the contribution of the target process and obtaining a more purified result; and secondly, some important implications, as follows. For chemical products, the change in the ratio of CE to C is helpful in evaluating the overall speed of the chemical processes; and the higher the proportion, the faster these processes might be. The contributions of C+CE play a significant role in stage 2 and 3, indicating that chemical reactions are more important in the most polluted period than in other periods. For the precursors (like $NH_3$), the smaller the value of CE, the scarcer they are, and reducing their emissions in that period would have the most efficient controlling effect on the air pollution. For example, when $SO_2$ is rich and $NH_3$ is deficient, the CE values of nitrate and ammonium are usually large and that of sulfate is small. This provides a standard for judging $NH_3$-rich or -poor periods. In addition, when EM or CE makes strong



positive contributions to PM$_{2.5}$, the suggestion is that additional benefits can be obtained by reducing PM$_{2.5}$ emissions at that
time. These implications can contribute to the formulation of refined emission reduction strategies.

The QDA method yields a strong general applicability and practical application prospects. Although the method was only

applied to PM$_{2.5}$, its components, and precursors in NAQPMS in this study, not only can it also be applied to any 3D
atmospheric chemistry model, but also to study any other pollutant. It can analyse the causes of pollution from different
substances. For example, application to the analysis of oxidants (e.g., O$_3$ and oxidative radicals), which are of wide concern in
CTMs, could enable in-depth studies of the nonlinear effects of chemical processes in the atmosphere. QDA can be used to
track the chain reactions caused by the changes in physical parameterization schemes or chemical reactions in CTMs, so as to
improve and test new mechanisms. Not only does this technique provide new reference ideas for the treatment of air pollution,
but it is also an important tool for further studying the formation processes of heavy pollution and the influence of different
physicochemical mechanisms.
**Code and data availability**
The observational data used in this study and the source codes of the QDA method are available online via ZENODO
(http://doi.org/10.5281/zenodo.5292895). Please contact Junhua Wang (wangjunhua@mail.iap.ac.cn) to obtain the model data
for the QDA method used in NAQPMS.
**Acknowledgements**
This work was funded by the National Natural Science Foundation of China (Grant No. 41877313). We thank the anonymous
reviewers for their constructive suggestions, which certainly helped to improve the manuscript.
**Author contributions**
Junhua Wang prepared the original data, designed and conducted the simulation, and carried out the QDA method. Baozhu Ge
and Xueshun Chen revised the paper and provided scientific guidance for the article design. Yayuan Dong gave advice on the
content of the article. Lei Kong provided help with the article framework and modified the model code. Yuanhang Zhang, Zifa
Wang, KeDing Lu, and Jie Li provided valuable suggestions for this article. Junhua Wang wrote the paper and all listed authors
have read and approved the final manuscript.
**Competing interests**
The authors declare that they have no conflicts of interest.

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





**Tables & Figures:**

**Table 1. Definitions of different factors considered in the QDA method**

| Notation | Equation | Definition |
|---|---|---|
| E | $\frac{\partial F}{\partial x_1}x_1 + \frac{1}{2!}\frac{\partial^2 F}{\partial x_1^2}x_1^2 + \frac{1}{3!}\frac{\partial^3 F}{\partial x_1^3}x_1^3 + \cdots$ | Pure emission contribution |
| M | $\frac{\partial F}{\partial x_2}x_2 + \frac{1}{2!}\frac{\partial^2 F}{\partial x_2^2}x_2^2 + \frac{1}{3!}\frac{\partial^3 F}{\partial x_2^3}x_2^3 + \cdots$ | Pure meteorology contribution |
| C | $\frac{\partial F}{\partial x_3}x_3 + \frac{1}{2!}\frac{\partial^2 F}{\partial x_3^2}x_3^2 + \frac{1}{3!}\frac{\partial^3 F}{\partial x_3^3}x_3^3 + \cdots$ | Pure chemistry contribution |
| ME | $\frac{1}{2!}\frac{2\partial^2 F}{\partial x_1 \partial x_2}x_1 x_2 + \frac{1}{3!}\sum_{a=1}^{2}\frac{3\partial^3 F}{\partial x_1^a \partial x_2^{3-a}}x_1^a x_2^{3-a} + \cdots$ | Interaction of meteorology and emissions |
| MC | $\frac{1}{2!}\frac{2\partial^2 F}{\partial x_2 \partial x_3}x_2 x_3 + \frac{1}{3!}\sum_{a=1}^{2}\frac{3\partial^3 F}{\partial x_2^a \partial x_3^{3-a}}x_2^a x_3^{3-a} + \cdots$ | Interaction of meteorology and chemistry |
| CE | $\frac{1}{2!}\frac{2\partial^2 F}{\partial x_1 \partial x_3}x_1 x_3 + \frac{1}{3!}\sum_{a=1}^{2}\frac{3\partial^3 F}{\partial x_1^a \partial x_3^{3-a}}x_1^a x_3^{3-a} + \cdots$ | Interaction of emissions and chemistry |
| MCE | $\frac{1}{3!}\left(\frac{\partial^3 F}{\partial x_1 \partial x_2 \partial x_3}6x_1 x_2 x_3\right) + \cdots$ | Three-way interaction of emissions, meteorology and chemistry |

Note: The order of capital letters under "Notation" does not represent the order of operators. For example, ME and EM can represent the same meaning, so it is uniformly expressed by ME in this paper.



**Table 2. Descriptions of the built-in scenario simulations in the QDA method**

|  | Simulation notation | Processes included in the simulations | Target values (e.g., model step of $t$ to $t+1$) |
|---|---|---|---|
| Base simulation | base | All model processes | $F(x_1, x_2, x_3)$, $F(0,0,0)$ |
| Built-in scenario simulations | $S_1$ | Only emission process | $F(x_1, 0, 0)$ |
|  | $S_2$ | Only meteorological process | $F(0, x_2, 0)$ |
|  | $S_3$ | Only chemical process | $F(0, 0, x_3)$ |
|  | $S_{13}$ | Emission and chemical processes | $F(x_1, 0, x_3)$ |
|  | $S_{23}$ | Meteorological and chemical processes | $F(0, x_2, x_3)$ |
|  | $S_{12}$ | Emission and meteorological processes | $F(x_1, x_2, 0)$ |

**Table 3. Descriptions of different processes considered in the IPR method**

| Description | Abbreviation |
|---|---|
| Emissions (local primary emissions in model) | emit |
| Horizontal advection | advhor |
| Vertical advection | advvert |
| Horizontal diffusion | difhor |
| Vertical diffusion | difvert |
| Wet deposition | wetdep |
| Dry deposition | drydep |
| Gas chemistry | gaschem |
| Inorganic aerosol chemistry | ISORR |
| Secondary aerosol chemistry | SOA |








**Table 4. Hourly average QDA results in different stages (unit: μg·m⁻³·h⁻¹)**

|  | Stage 1 | | Stage 2 | | Stage 3 | | Stage 4 | |
|---|---|---|---|---|---|---|---|---|
| **Hourly change** | −1.36 | | 1.79 | | 0.1 | | −11.84 | |
| **M** | −2.60 | 191.18% | 0.34 | 18.99% | −1.75 | −1750.00% | −12.62 | 106.59% |
| **E** | 0.88 | −64.71% | 0.82 | 45.81% | 0.88 | 880.00% | 0.63 | −5.32% |
| **C** | 0.31 | −22.79% | 0.37 | 20.67% | 0.23 | 230.00% | −0.08 | 0.68% |
| **COUP** | 0.05 | −3.68% | 0.26 | 14.53% | 0.74 | 740.00% | 0.23 | −1.94% |
| **EM** | −0.08 | 5.88% | −0.08 | −4.47% | −0.09 | −90.00% | −0.11 | 0.93% |
| **CE** | 0.10 | −7.35% | 0.13 | 7.26% | 0.67 | 670.00% | 0.43 | −3.63% |
| **MC** | −0.01 | 0.74% | 0.20 | 11.17% | 0.03 | 30.00% | −0.14 | 1.18% |
| **MCE** | 0.04 | −2.94% | 0.01 | 0.56% | 0.13 | 130.00% | 0.05 | −0.42% |

Note: Hourly change=M+C+E+COUP; COUP=EM+CE+MC+MCE.

**Table 5. Hourly average IPR results in different stages (unit: μg·m⁻³·h⁻¹)**

|  | emit | advhor | advvert | difhor | difvert | gaschem | drydep | ISORR | wetdep | SOA |
|---|---|---|---|---|---|---|---|---|---|---|
| **Stage 1** | 0.88 | −3.32 | 0.94 | −0.002 | −0.28 | 0.00 | −0.02 | 0.44 | 0.00 | 0.02 |
| **Stage 2** | 0.82 | 0.03 | 0.58 | −0.01 | −0.34 | 0.00 | −0.03 | 0.71 | 0.00 | 0.02 |
| **Stage 3** | 0.88 | 0.18 | −1.45 | −0.01 | −0.57 | 0.00 | −0.04 | 1.07 | 0.00 | 0.04 |
| **Stage 4** | 0.63 | −13.26 | 0.71 | −0.002 | −0.22 | 0.00 | −0.03 | 0.32 | 0.00 | 0.003 |




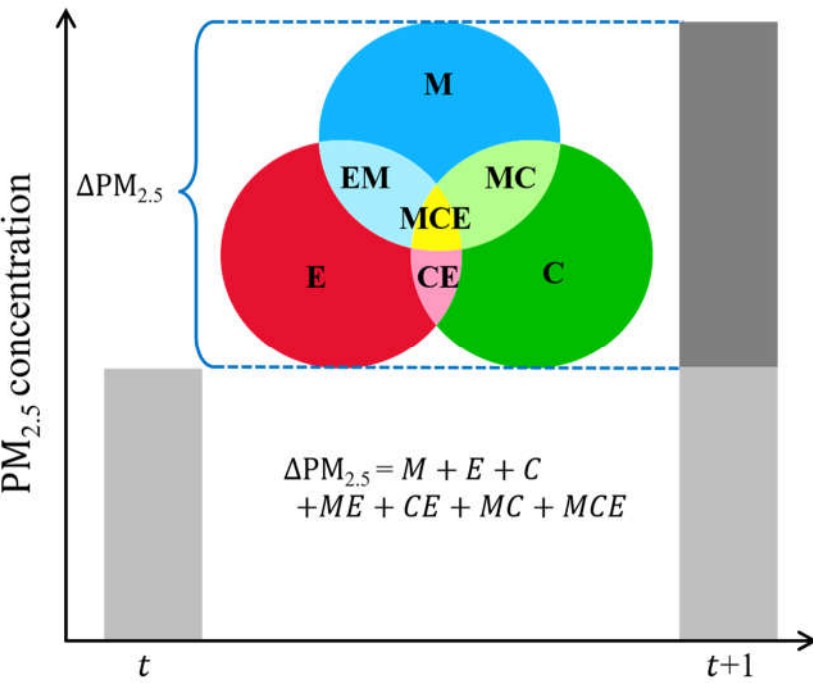


**Figure 1. Graph theory of the QDA method. The total area of the colour graphics represents the hourly change in the PM$_{2.5}$
concentration between *t* and *t*+1, which can be resolved into seven quantitative analytical factors—see Table 1 for meanings of the
abbreviations.**





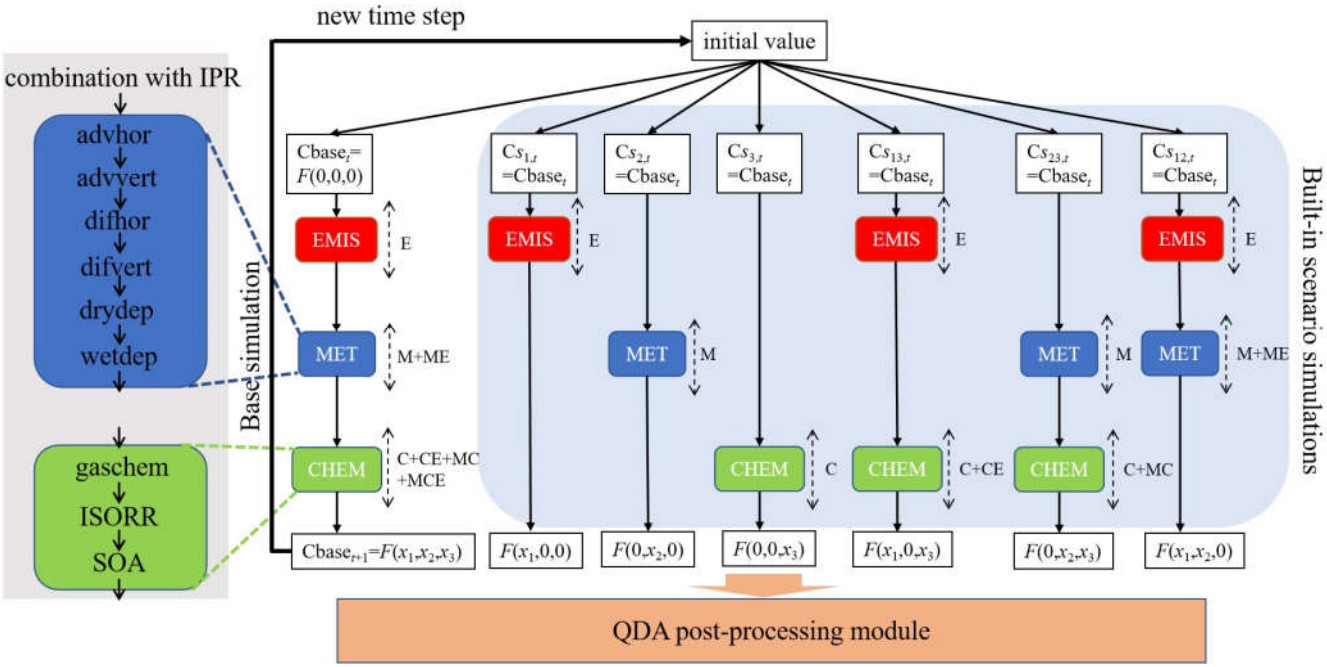

**Figure 2. Flow chart of the QDA method [see Eqs. (15)–(24) in Sect. 2.3 for the QDA post-processing module].**



Figure 3. (a) Model domain and (b) observation sites in Beijing for the evaluation in this study.



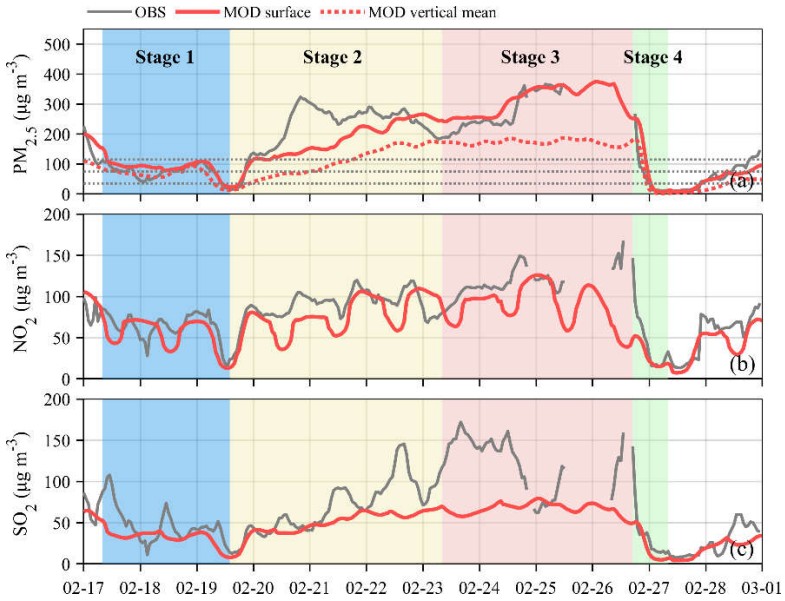


**Figure 4. Observations (OBS) and simulation results (MOD) for (a) PM$_{2.5}$, (b) NO$_2$ and (c) SO$_2$ in Beijing. All simulation and observation results are averaged over the Beijing area. The three grey dotted lines indicate 35, 75 and 115 µg·m$^{-3}$.**















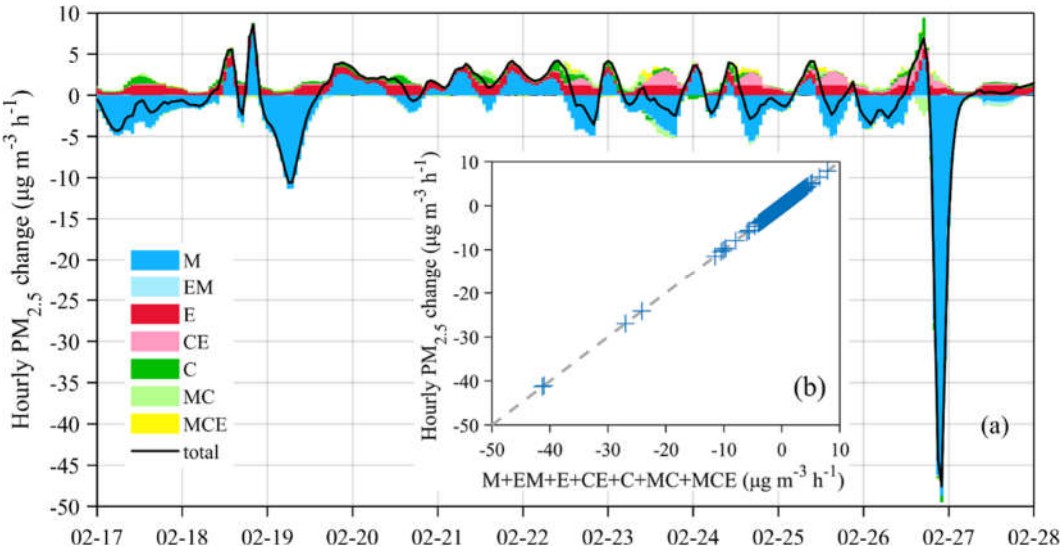

**Figure 5. (a) QDA results and PM2.5 hourly concentration change (black line) between adjacent hours and (b) scatterplot of the sum of all contributions versus the PM2.5 hourly concentration change. The scattered points all fall on the 1:1 diagonal line, indicating that the PM2.5 concentration change can be fully resolved by the QDA results.**





**Figure 6. Vertical mean hourly contribution of each factor in (a–d) QDA and (e–h) its IPR results that influence the hourly mean PM$_{2.5}$ change within the model height in different stages. There is correspondence between the upper and lower subgraphs and the bar values are available in Tables 4 and 5. Taking the M bar in (a) for example, M is composed of six contributing parts as displayed in the M bar of (e): 'advhor', 'advvert', 'difhor', 'difvert', 'drydep' and 'wetdep', respectively.**





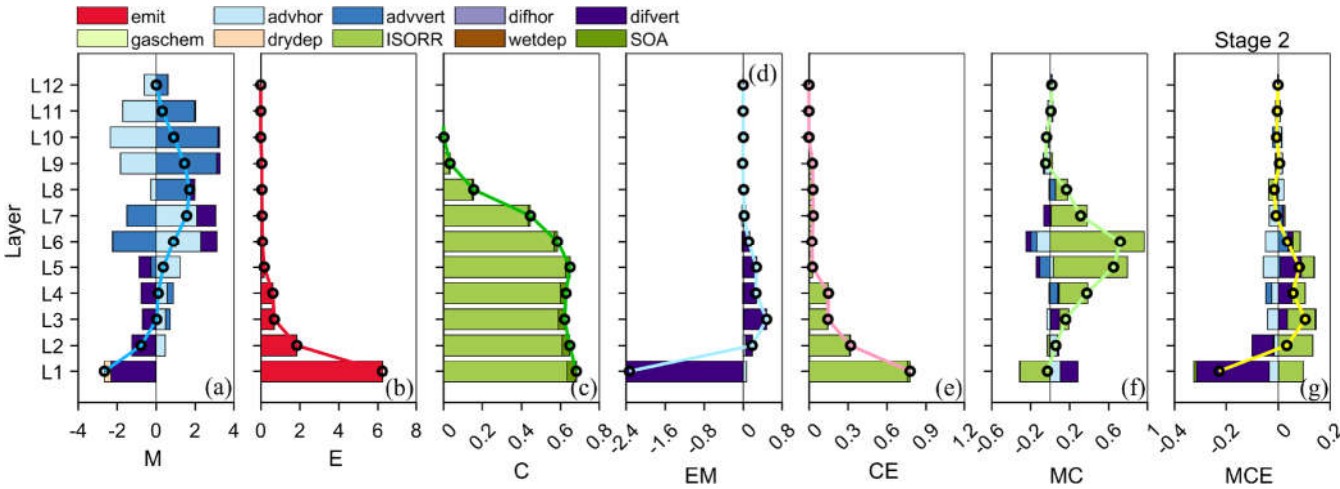

**Figure 7. Vertical process decomposition of the QDA results in stage 2 (the black arrow and coloured lines indicate the average change in the PM$_{2.5}$ concentration, and the results for other stages are shown in Figs. S7–S9; unit: μg·m$^{-3}$·h$^{-1}$). The layer heights, L1–L12 are: 112, 222, 361, 531, 740, 989, 1279, 1627, 2046, 2555, 3163, and 3890 m.**





**Figure 8. QDA results for (a–d) NO$_x$, (e–h) SO$_2$, (i–l) NH$_3$, (m–p) nitrate, (q–t) sulfate, and (u–x) ammonium, during different stages of the episode.**



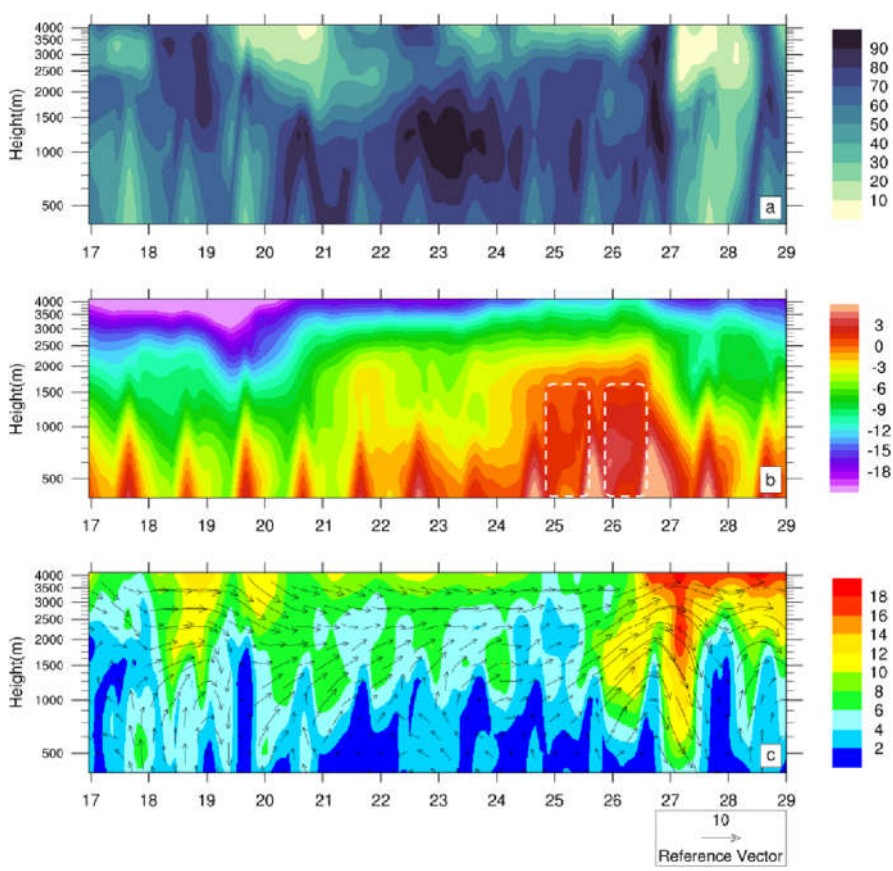


**Figure 9. Vertical distribution of (a) RH, (b) temperature and (c) wind field from 17 to 28 February 2014 over the Beijing area in a**
**sigma-*p* vertical coordinate. The white dotted frames in (b) represent a temperature inversion. The vector diagram in (c) represents**
**the horizontal wind field, and the shading denotes the wind speed.**











