# Peer review of "A quantitative decoupling analysis (QDA v1.0) method for assessing 1 the contributions of meteorology, emissions, and chemistry to fine 2 particulate pollution 3"

_Geoscientific Model Development, 2023_

## Author Comment (AC1)

**Response to Referee #1 (GMD-2023-22)**

We Thank Reviewer for his/her constructive comments.

Responses to the comments:

**Comments**: This resubmitted manuscript (ID: GMD-2023-22) has been well improved from the previous version (ID: GMD-2021-259), especially in terms of the explanation of the QDA method. Now we can clarify the differences and/or relationship between the QDA method and previous methods such as SAA, FS, and IPR. However, I have fundamental questions about the combination of QDA and IPR as described in Section 2.2. In my understanding, the component of "wetdep" in IPR (I believe this term is corresponded to "CLDS" in CMAQ's IPR; https://www.cmascenter.org/cmaq/science_documentation/pdf/ch16.pdf) includes wet deposition process and the aqueous-phase chemistry which is highly important in the sulfate aerosol production. If so, it cannot be separated as Eqs. (26) and (27), because "wetdep" term can be attributed both in M and C in QDA. As replied in the previous review process, the aqueous-phase chemistry is included in the "gaschem" in this case, right? The current manuscript is still not clear regarding this point. If the readers know CMAQ IPR, the current description will lead to confusion. In addition, as listed in Table 5, "gaschem" was also continuously zero through this analyzed period. Did this stand for no production via the aqueous-phase chemistry in all four stages? In this case, what is the pathway of sulfate aerosol? The term "ISORR" seems to be the main component of C in QDA; however, I do not follow why the sulfate production is attributed to "ISORR". At the current quality, it is required for furthermore revisions in Section 2.2 and Table 3 to point out what stands for each IPR component, and a more in-depth discussion of the production process.
**Reply:** Thanks for this comment. I would reply in the following parts.

**Comment 1:** For the question about the "wetdep" cannot be separated as Eqs. (26) and (27), because "wetdep" term can be attributed both in M and C in QDA.
**Reply:** The component of "wetdep" in CMAQ's IPR includes wet deposition process and the aqueous-phase chemistry, while the component of "wetdep" in NAQPMS's IPR only includes wet deposition process, so "wetdep" in our study can only contribute in M.

**Comment 2:** As replied in the previous review process, the aqueous-phase chemistry is included in the "gaschem" in this case, right? The current manuscript is still not clear regarding this point. If the readers know CMAQ IPR, the current description will lead to confusion.
**Reply:** We feel sorry that we did not provide enough description on the difference between CAMQ's IPR and NAQPMS's IPR and we have revised the manuscript accordingly. In the revised manuscript, this issue has been supplemented in section 2.2 as: "It should be noted that the aqueous-phase chemistry is calculated in the gas chemistry module ("gaschem" in Table 3) of NAQPMS, while the aqueous-phase chemistry of CMAQ is calculated in the wet deposition module, which may lead to different results of IPR in different models."
**Changes in the manuscript: lines 286-288.**

**Comment 3:** In addition, as listed in Table 5, "gaschem" was also continuously zero through

this analyzed period. Did this stand for no production via the aqueous-phase chemistry in all four stages? In this case, what is the pathway of sulfate aerosol? The term "ISORR" seems to be the main component of C in QDA; however, I do not follow why the sulfate production is attributed to "ISORR". At the current quality, it is required for furthermore revisions in Section 2.2 and Table 3 to point out what stands for each IPR component, and a more in-depth discussion of the production process.

**Reply:** In Table 5, "gaschem" was also continuously zero because what we analyzed in this study is the quantity of $PM_{2.5}$ produced directly by each process. For example, aqueous-phase chemistry in "gaschem" produce liquid sulfuric acid (this step would not increase the amount of $PM_{2.5}$), and these liquid sulfuric acid should then undergo gas-particle partitioning to produce sulfate particles (this step would increase the amount of $PM_{2.5}$). In NAQPMS model, the gas-particle partitioning process is included in "ISORR" while the aqueous-phase chemistry is included in "gaschem", which make the term "ISORR" the main component of C in QDA. We have modified the description of the process in section 2.2 and Table 3.

**Changes in the manuscript: lines 425-430, lines 1000-1002.**

---

## Author Comment (AC2)

**Response to Referee #2 (GMD-2023-22)**

We Thank Reviewer for his/her constructive comments.

Responses to the comments:

**Comments**: The paper is quite ambitious, as it presents a novel approach (QDA - quantitative decoupling analysis) to quantify the effects of emissions, meteorology, chemical reactions and nonlinear interactions on $PM_{2.5}$ concentrations.

   As the authors correctly state, in literature there are already existing approaches to perform this task, as IPR (integrated process rate), SAA (scenario analysis approach), FS (factor separation). On this last topic ... I have two main concerns on the current version of the paper **Reply:** Thanks for this comment. I would reply in the following parts.

**Comment 1:** focus on pros and cons of the different approaches. Even if the authors provide some hints of the pros and cons of the approaches (section 2.1.4) still it is not clear to me why we need another approach (QDA) and why the existing ones are not sufficient. Please better explain this, and also provide a more schematic and syntetic view of pros and cons of the different approaches, i.e. also with a table or graphical view.

   **Reply:** Thanks for this important comment. The differences between the QDA method and other mainstream methods in analyzing the effects of meteorology, chemistry and emission on the $PM_{2.5}$ are explained theoretically in Sect. 2.1.4. The gain of the QDA method compared to the IPR is straightforward that mainly exists in the resolving of the nonlinear effects among the different processes. As we illustrated in Sect. 3.3 and 3.4, we can resolve the changes of $PM_{2.5}$ into the pure effects of different processes (i.e. M, E, C) as well as their nonlinear interactions (i.e., EM, MC, CE and MCE) by using the QDA method, while the changes of $PM_{2.5}$ are only attributed to the effects of meteorology, chemistry and emission in the IPR method. The inclusion of the nonlinear effects in the QDA method, on the one hand, can overcome the problems of the no uniqueness of the results of IPR method which is dependent on the sequence of the different processes due to the nonlinearity of the atmospheric chemistry. On the other hand, it can provide us with more information on the state of atmosphere chemistry by analyzing the nonlinear effects. For example, the analysis of the EM results of the ammonium (Sect. 3.5.1) can help us identify whether there is a $NH_3$-rich or $NH_3$-poor condition.

   The FS method is instructive to the development of the QDA method. However, in previous studies the FS method was mainly used in the analysis of the effects of model input parameters, such as the emission inventory and topography (Alpert et al., 1999; Tao et al., 2005), or the effects of specific physical variables, such as the synoptic-scale wind and the atmospheric moisture (Rotstein et al., 2021), which has not been used in the model processes. The QDA method takes the FS method as a reference and applies it to the analysis of the effects of meteorology, emission, chemistry as well as their interactions on the $PM_{2.5}$ concentrations. From this perspective, the QDA method is similar to the FS method but with different analysis objects and is applied within model steps.

   The SAA has more than one choice of operation path, which leads to great uncertainty in the result. QDA method can overcome the problems of the SAA method in the dependence of the choices of fixed emission, which could yield consistent results of the effects of

meteorological and emission changes on the PM$_{2.5}$ variations.
**Changes in the manuscript: lines 559-578.**

**Comment 2:** also, I would like to see not only theoretically, but also in practice, what you gain using QDA instead of using other approaches. To do so, please apply, on the same data and episode, also other approaches (i.e. SAA, FS ...) to see if you really gain (and what you gain) on the results' quality and interpretation, using the QDA approach.

**Reply:** Thanks for this comment. Because FS may cause memory overflow or simulation error when it is used to calculate meteorological and chemical actions (turn off meteorology or chemistry for long-term simulation), only the differences between SAA and QDA are analyzed with a specific example in the revised manuscript.

To investigate the gains of the QDA method with regards to the SAA method, the SAA method was applied to the same cases in this study and compared with the QDA method. Commonly, the SAA method compares two cases that started at different times but lasted for the same duration. The concentration differences between these two cases can be divided into anthropogenic contribution and meteorological contribution. By keeping the emissions unchanged and changing the meteorological field in simulation, the contribution of meteorological changes to the PM$_{2.5}$ can be calculated, and the remaining change is the impact caused by the emissions. However, the results of this method are dependent on the choice of the emission. The different choices usually change the results of the SAA. In order to evaluate the proportion of anthropogenic and meteorological contributions in 19–27 February 2015 compared to the same period in 2014, the following scenarios are designed according to the SAA method (Table R1).

The changes in the concentrations of PM$_{2.5}$ between the two cases can be expressed as: PM$_{2.5\_2015base}$ − PM$_{2.5\_2014base}$, and there are two paths to calculate the contribution of changes in emission and meteorological fields based on the SAA method:

Path 1: According to the Table R1, we can do it in these two steps:
$$\text{2014base} \rightarrow \text{2015\_emis2014} \rightarrow \text{2015base}$$
The concentration changes for the two steps are:
$$\Delta\text{MET1} = \text{PM}_{2.5\_2015\_emis2014} - \text{PM}_{2.5\_2014base}$$
$$\Delta\text{ANT1} = \text{PM}_{2.5\_2015base} - \text{PM}_{2.5\_2015\_emis2014}$$

$\Delta$MET1 represents the amount of PM$_{2.5}$ concentration change due to meteorological changes, and $\Delta$ANT1 represents the anthropogenic contribution due to the emission changes.

Path 2: According to the Table R1, we can also do it in these two steps:
$$\text{2014base} \rightarrow \text{2014\_emis2015} \rightarrow \text{2015base}$$
$$\Delta\text{ANT2} = \text{PM}_{2.5\_2014\_emis2015} - \text{PM}_{2.5\_2014base}$$
$$\Delta\text{MET2} = \text{PM}_{2.5\_2015base} - \text{PM}_{2.5\_2014\_emis2015}$$

In Table R3, the concentration differences can be decomposed into the contributions of anthropogenic ($\Delta$ANT), meteorological ($\Delta$MET), and nonlinear effects ($\Delta$COUP), the sum of these three components represents the total change in PM$_{2.5}$ concentrations of 2015 compared to the same period of 2014. $\Delta$MET is equal to the M mentioned above, $\Delta$ANT contains C, E and CE, and the rest of the nonlinear effects are classified as $\Delta$COUP. $\Delta$COUP represents the nonlinear interaction between meteorological and anthropogenic effects and it can also be used as a range of uncertainty in pure meteorological and anthropogenic contributions. In

stage 2 and stage 3, reduction in PM$_{2.5}$ concentrations occurred in 2015 is more significant compared to 2014, and changes in meteorological conditions mainly contributed to it. Emissions in 2015 are lower than in 2014, which is the reason why ΔANT was negative in all stages, and the reduction in emissions will not only lead to the reduction in the direct contribution to PM$_{2.5}$, but also to the reduction in the precursors of chemical reactions, which will result in the less generation of secondary aerosol.

ΔMET2 and ΔANT2 represent the changes in PM$_{2.5}$ concentration due to meteorological changes and anthropogenic changes, respectively. And there is ΔMET1+ΔANT1 = ΔMET2+ΔANT2. It can be seen in the Table R2 that the results of anthropogenic and meteorological contributions of PM$_{2.5}$ obtained by different paths are not consistent, indicating that the results obtained by SAA have a large uncertainty. In stage 3, ΔMET1 valued -6.27 µg m$^{-3}$ indicates that meteorological fields in 2015 were more unfavourable for PM$_{2.5}$ increases than in 2014, while ΔMET2 valued 0.3 µg m$^{-3}$ reflects that the meteorological fields in 2015 were more favourable for PM$_{2.5}$ increases. The meteorological contributions of the two paths are even opposite. The anthropogenic contributions obtained by the two paths are also opposite in sign in stage 1 and 3, reflecting the uncertainties in the SAA method.

In Table R3, the concentration differences can be uniquely decomposed into the contributions of anthropogenic (ΔANT), meteorological (ΔMET), and nonlinear effects (ΔCOUP), the sum of these three components represents the total change in PM$_{2.5}$ concentrations of 2015 compared to the same period of 2014. ΔMET is equal to the M mentioned above, ΔANT contains C, E and CE, and the rest of the nonlinear effects are classified as ΔCOUP. ΔCOUP represents the nonlinear interaction between meteorological and anthropogenic effects and it can also be used as a range of uncertainty in pure meteorological and anthropogenic contributions. In stage 2 and stage 3, reduction in PM$_{2.5}$ concentrations occurred in 2015 is more significant compared to 2014, and changes in meteorological conditions mainly contributed to it. Emissions in 2015 are lower than in 2014, which is the reason why ΔANT was negative in all stages, and the reduction in emissions will not only lead to the reduction in primary PM$_{2.5}$, but also to the reduction in the precursors of chemical reactions, which will result in smaller formation of secondary aerosol.

In 2015 case, the difference between meteorological and anthropogenic contributions within the same stage is smaller than in 2014, so the variation in PM$_{2.5}$ is smaller in 2015. The meteorological fields can always provide clearance timely, so that the pollution produced by anthropogenic action cannot accumulate continuously, which is the main reason why there is no persistent heavy haze in 19–27 February 2015. Therefore, these results suggest that the QDA method can overcome the problems of the SAA method in the dependence of the choices of fixed emission, which could yield consistent results of the effects of meteorological and emission changes on the PM$_{2.5}$ variations.

**Table R1. List of the scenario simulations**

| Scenario name | Time of meteorological field | Time of emission inventory |
| --- | --- | --- |

| | | |
|---|---|---|
| 2014base | 19–27 February 2014 | 19–27 February 2014 |
| 2015base | 19–27 February 2015 | 19–27 February 2015 |
| 2014_emis2015 | 19–27 February 2014 | 19–27 February 2015 |
| 2015_emis2014 | 19–27 February 2015 | 19–27 February 2014 |

**Table R2. Results of SAA for comparing vertical mean hourly concentration change of PM$_{2.5}$ in 19–27 February 2015 with the same period in 2014**

| | Path 1 | | Path 2 | |
|---|---|---|---|---|
| Stage | $\Delta$MET1 ($\mu$g m$^{-3}$) | $\Delta$ANT1 ($\mu$g m$^{-3}$) | $\Delta$MET2 ($\mu$g m$^{-3}$) | $\Delta$ANT2 ($\mu$g m$^{-3}$) |
| Stage 1 | 142.95 | -0.45 | 138.96 | 3.55 |
| Stage 2 | -204.00 | 0.31 | -217.08 | 13.39 |
| Stage 3 | -6.27 | 0.63 | 0.30 | -5.94 |
| Stage 4 | 167.50 | -0.68 | 174.02 | -7.20 |

**Table R3. Results of QDA for comparing vertical mean hourly concentration change of PM$_{2.5}$ in 19–27 February 2015 with the same period in 2014**

| Stage | $\Delta$MET ($\mu$g m$^{-3}$) | $\Delta$ANT ($\mu$g m$^{-3}$) | $\Delta$COUP ($\mu$g m$^{-3}$) |
|---|---|---|---|
| Stage 1 | 160.64 | -7.25 | -7.19 |
| Stage 2 | -178.10 | -13.08 | -13.42 |
| Stage 3 | 58.89 | -43.55 | -20.63 |
| Stage 4 | 166.17 | -2.70 | 3.04 |

Notes: $\Delta$MET represents changes in $PM_{2.5}$ concentrations due to changes in emission inventories, and also including changes in chemical reactions due to changes in emissions, $\Delta$ANT represents the change of $PM_{2.5}$ concentration caused by the change of meteorological fileds, $\Delta$COUP represents the change of $PM_{2.5}$ concentration caused by nonlinear effects.

**Changes in the manuscript: lines 579-634.**

**Comment 3:**

Minor comment
please move part of the Equations (section 2) in the supplementary material, so that the main concepts you propose remain in the main part of the manuscript, and the more technical part goes in the Annex.

**Reply:** Thanks for this comment. Part of equations in section 2 have been moved to the supplementary material in the revised manuscript as Eqs. (S1)–(S11).

**Changes in the manuscript: lines 227-228, lines 294-295, lines 296-299.**

**References:**

Alpert, P., Tsidulko, M., and Itzigsohn, D.: A shallow, short-lived meso-beta cyclone over the Gulf of Antalya, eastern Mediterranean, Tellus Ser. A-Dyn. Meteorol. Oceanol., 51, 249-262, 10.1034/j.1600-0870.1999.t01-2-00006.x, 1999.

Rotstein, M., Alpert, P., and Rostkier-Edelstein, D.: A Factor Separation Study of the Effect of Synoptic-Scale Wind, Atmospheric Moisture and of Their Synergy on the Diurnal Temperature Range During the Israeli Summer, J. Geophys. Res.-Atmos., 126, 18, 10.1029/2021jd034923, 2021.

Tao, Z. N., Larson, S. M., Williams, A., Caughey, M., and Wuebbles, D. J.: Area, mobile, and point source contributions to ground level ozone: a summer simulation across the continental USA, Atmos. Environ., 39, 1869-1877, 10.1016/j.atmosenv.2004.12.001, 2005.

---

## Author Comment (AC3)

**Response to executive editor (gmd-2023-22)**

**Comment 1:** Unfortunately, after checking your manuscript, it has come to our attention that it does not comply with our "Code and Data Policy".

https://www.geoscientific-model-development.net/policies/code_and_data_policy.html

In your manuscript, you state that to get access to the model data for the QDA method in NAQPMS, it is necessary to contact you. This is not acceptable according to our policy, and your manuscript is currently in an irregular situation, as it should have not been accepted for Discussions with such shortcomings. You must publish all the code and data used to carry on your work in one of the acceptable repositories stated in our policy.

Also, in your manuscript, you state that you use several other models and datasets, and you have not published their code or even mentioned them in the Code Availability Section. These include CMAQ 4.6, the MEIC data, fields from WRF, etc. Even the link that you provide for MEIC data in the text is broken and the linked webpage does not serve the data. Also, the Zenodo repository that you include with your manuscript does not include an explanation about what is each file, and the names of them do not help.

Moreover, some data in the Zenodo repository is in .xlsx format, which depends on proprietary software to get correct access to it. I recommend you to save and publish such files in OpenDocumentFormat (.ods), which is an ISO standard and relies on free software.

Therefore, please, publish the requested data and code in one of the appropriate repositories, and reply to this comment with the relevant information (link and DOI) as soon as possible, as it should be available for the Discussions stage.

Also, you must include in a potentially reviewed version of your manuscript the modified 'Code and Data Availability' section, with the DOI of the new repositories.

Please, note that if you do not fix this problem, we will have to reject your manuscript for publication in our journal.

**Reply:** We feel sorry for the noncompliance with the "code and data" policy of the GMD. We have submitted all the codes related to the QDA method and the observation data that used to validate the QDA method in our original submission. Following the requirement of the editor, we have uploaded the emission dataset obtained from the MEIC and the WRF fields in the new repository. Note that the source code of the CMAQ 4.6 has been included in the source code we submitted to the repositories. A

document explaining the meaning of each file is also added in the new repositories and the xlsx files in the original repository have been transferred to the OpenDocumentFormat (.ods). The DOI of the new repositories is

10.5281/zenodo.8217774 and the 'Code and Data Availability' section has been modified as:

The observational data used in this study and the source codes of the QDA method are available online via ZENODO (http://doi.org/10.5281/zenodo.8217774). The emission data obtained from MEIC (http://meicmodel.org.cn/?page_id=1772&lang=en) and 24-hour WRF simulation data that were used to drive the QDA method are also included.